# Skip Connections and Generalization: A PAC-Bayesian Perspective

## Abstract

With the growing popularity of large-scale models, neural networks with massive numbers of parameters and increasingly complex architectures have been widely deployed in practice. While significant theoretical efforts have been devoted to understanding generalization in the overparameterized regime, the role of non-parametric architectural structures remains less well understood. In this paper, we study the structural influence of skip connections on generalization through the lens of the PAC-Bayesian framework. We introduce a notion of general weight correlation to formally capture inter-layer dependencies induced by skip connections. Based on this framework, we theoretically show that correlations between adjacent layers hinder generalization, thereby explaining why ResNet-style skip connections provide an advantage. We further analyze the interaction between cross-layer and intra-layer correlations and prove that heterogeneous correlation structures across layers promote generalization. Finally, we empirically validate our framework on all skip-connection configurations in multilayer perceptrons and convolutional networks, demonstrating that our approach effectively isolates the contribution of skip connections to generalization.

## 1 Introduction

With the rapid growth of computational resources, neural networks with increasingly large parameter counts have become ubiquitous across diverse application domains. Beyond sheer model size, architectural innovations have also been a key driver of progress Xu et al. (2024). Among these, skip connections have emerged as a fundamental component of modern deep networks since their introduction in ResNet. By introducing direct links across layers, skip connections not only stabilize training but also enhance generalization performance. A substantial body of work has sought to explain these benefits. However, most existing studies approach the problem from a single perspective—such as optimization Li et al. (2018), algorithmic stability Hardt et al. (2016), or the Neural Tangent Kernel (NTK) Arora et al. (2019)—and typically restrict their analysis to one specific form of skip connection.

Fig. 1 illustrates the correlation matrices of posterior weights for different skip-connection configurations in a 5-layer MLP. The posterior distribution is obtained by applying small-learning-rate perturbations. Notably, the correlation structure of the weights changes substantially even with minimal architectural modifications—for instance, adding a single skip connection at the second layer (Fig. 1b) or removing one connection (Fig. 1c). These results suggest that skip connections strongly influence cross-layer dependencies captured in the posterior. Such sensitivity provides a natural entry point for PAC-Bayesian analysis, which explicitly links posterior correlations to generalization. From this perspective, we can theoretically characterize how generalization varies with different skip-connection patterns, offering principled guidance for designing non-parametric architectures.

However, even for a toy MLP with fewer than 500 parameters, approximating the full correlation matrix requires at least 2,000 runs (roughly four times the number of parameters) to obtain a reasonable estimate. This quickly becomes infeasible for modern neural networks with billions of parameters. Inspired by Laplace approximation of Hessian matrices (Ritter et al., 2018), we factorize the correlation matrix by using the Kronecker product. Our approach naturally extends prior work on weight correlation (Jin et al., 2020) and weight volume (Jin et al., 2022), both of which focus only on intra-layer correlations while treating layers independently. In contrast, skip connections inherently

induce dependencies across layers. To capture this effect, we introduce the notion of general weight correlation, which models inter-layer dependencies, and propose a correlation matrix $R$ to explicitly represent the influence of skip connections (see Fig. 2). We then provide a theoretical analysis of how different structures of $R$ affect generalization, thereby explaining the discrepancies observed across different types of skip connections. To validate our framework, we conduct experiments on MLPs with Fashion-MNIST and CNNs with CIFAR-10. We evaluate our method using Kendall's $\tau$ correlation coefficient Kendall (1938) and demonstrate its ability to effectively capture the role of skip connections.

Our main contributions are summarized as follows:

- To the best of our knowledge, this is the first work to analyze the non-parametric structural influence of skip connections on generalization gaps from a PAC-Bayesian perspective. We introduce the concept of general weight correlation to capture inter-layer dependencies induced by skip connections.

- Within this framework, we theoretically prove that correlations between adjacent layers impede generalization, thereby explaining the generalization advantage of ResNet-style skip connections.

- We further show how cross-layer weight correlations interact with intra-layer correlations under the setting of homogeneous cross-layer dependence. Our analysis reveals that generalization benefits from heterogeneous (layer-specific) correlation structures.

- We empirically validate our framework on all possible skip-connection configurations in 5-layer MLPs and CNNs. The results demonstrate that our method effectively captures the influence of skip connections, isolating their contribution to generalization.

## 2 RELATED WORK

### 2.1 PAC-BAYES GENERALIZATION BOUNDS

Classical PAC-Bayesian analyses bound the true risk of a Gibbs or posterior-averaged predictor by balancing the empirical risk with a complexity term measured by a Kullback-Leibler divergence between a posterior over hypotheses and a prior (McAllester, 1999; Langford et al., 2001; Catoni, 2007). These early works established data-independent priors, generic KL penalties, and temperature-style trade-offs that remain the backbone of modern formulations. Recent work adapts these ideas to deep networks and stochastic training pipelines. Dziugaite & Roy (2017; 2018) construct non-vacuous, data-dependent bounds for overparameterized nets by optimizing the posterior and sometimes the prior subject to PAC-Bayes constraints. Margin information has been incorporated to tighten the empirical term and connect PAC-Bayes to classical margin theory (Neyshabur, 2017). Other directions study how SGD implicitly induces "flat" posteriors or noise-averaging effects that PA–Bayes can capture through perturbation-sensitive priors/posteriors and training-time noise models (Letarte et al., 2019). Complementary threads relate PAC-Bayes to norm- or compression-based capacities, spectral controls, and sharpness-style surrogates, yielding bounds that move with optimization geometry rather than parameter count. However, most works focus on overall generalization but do not analyze cross-layer parameter correlations or the role of skip connections.

Flatness-based generalization measures typically rely on estimating the Hessian or Fisher Information Matrix and relate reduced sharpness of the loss landscape to improved generalization (Zhang et al., 2025). These approaches emphasize local geometric properties—such as curvature, layer-wise sensitivity, or perturbation stability—but largely overlook interactions across layers. In contrast, our work focuses on cross-layer parameter correlations, offering a measure that clarifies how skip connections influence generalization.

Information-theoretic analyses (Xu & Raginsky, 2017) instead study mutual information between data and parameters, providing a global characterization of information flow in learning. Our approach differs in that we examine the correlation structure among weights themselves, motivated by structured dependencies induced by architectural elements such as skip connections. For this purpose, McAllester's PAC-Bayesian framework offers a simple and interpretable tool for connecting such structural correlations to generalization behavior.

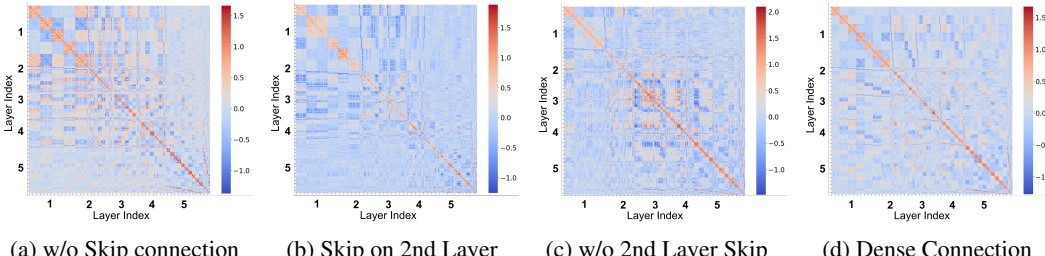

(a) w/o Skip connection        (b) Skip on 2nd Layer        (c) w/o 2nd Layer Skip        (d) Dense Connection

Figure 1: **Full Correlation Matrices of Posterior Weights with Different Skip Connection Patterns.** We train a toy 5-layer MLP with parameters less than 500 to show the full Correlation Matrices. We sample the posterior samples of weight by SWAG Maddox et al. (2019) which is scalable method to conduct uncertainty estimation. It is achieved by perturbation around the local minima with a small learning rate on loss surface. Details about sampling is shown in Appendix A. The first figure corresponds to the MLP without skip connections. The second establishes connections only at the 2nd layers. The third figure includes skip connections that exclude only the 2nd layer, and the last one shows the dense MLP.

## 2.2 FLAT MINIMA AND GENERALIZATION

The connection between the geometry of the loss landscape and generalization has been studied for several decades. Early work by Hochreiter & Schmidhuber (1997) introduced the idea that flat minima, where the loss remains nearly constant under small perturbations of the parameters, are strongly associated with improved generalization. Their argument, grounded in a Minimum Description Length (MDL) perspective, suggested that flatness reflects the robustness of the learned solution. Subsequent empirical and theoretical studies reinforced this principle. Keskar & Socher (2017) provided evidence that sharp minima often correspond to poor generalization, particularly when training with large-batch methods. Li et al. (2018) developed visualization tools to illustrate how optimization trajectories converge to regions of varying sharpness, offering geometric intuition for the flatness–generalization link. Jiang et al. (2019) further connected margin-based generalization to flatness, showing that flatter minima correlate with wider margins and tighter generalization bounds. While the flatness perspective provides a compelling explanation of generalization, existing work largely treats it as an isolated principle, without integration into PAC-Bayesian frameworks or explicit consideration of architectural mechanisms such as skip connections.

## 2.3 SKIP CONNECTIONS AND THEORETICAL ANALYSIS

Skip connections, first popularized by residual networks He et al. (2016), are widely recognized for their empirical benefits in stabilizing optimization and enabling the training of very deep models. From the optimization perspective, theoretical studies have demonstrated that residual links reshape the loss landscape to reduce sharpness and ease convergence. Zhang et al. (2019) provided early evidence that skip connections facilitate gradient flow and mitigate vanishing or exploding gradients, while Li et al. (2021) analyzed how skip connections ease optimization and improve gradient flow. These works frame skip connections primarily as a tool for optimization stability rather than for generalization guarantees. A complementary line of research examines the role of parameter correlations induced by modern architectures. Jin et al. (2020) studied how weight correlation within layers affects generalization and illustrated that correlated parameters can implicitly constrain hypothesis complexity and lead to sharper theoretical bounds, while the cross-layer correlations that are naturally amplified by skip connections due to the direct reuse of features and gradients across depth remain unexplored. Existing work analyses primarily account for the optimization benefits of skip connections, while their impact on generalization remains largely unexplored, with no prior work employing PAC-Bayesian theory to study them.

## 3 PRELIMINARY

Our analysis is based on supervised classification. Let $\boldsymbol{x} \in \mathcal{X} \subseteq \mathbb{R}^d$ denote the input, $y \in \mathcal{Y} = \{1, \ldots, \kappa\}$ the label, and $D$ the unknown data distribution over $\mathcal{X} \times \mathcal{Y}$. A hypothesis $h \in \mathcal{H}$ maps inputs to predictions in $[0,1]^\kappa$. Given i.i.d. samples $S = \{(\boldsymbol{x}_i, y_i)\}_{i=1}^n \sim D^n$ and a loss function $\ell : [0,1]^\kappa \times \mathcal{Y} \to \mathbb{R}^+$, we denote natural and empirical risks as

$$R(h) = \mathbb{E}_{(\boldsymbol{x},y) \sim D}\left[\ell(h(\boldsymbol{x}), y)\right], \qquad \widehat{R}(h) = \frac{1}{n} \sum_{i=1}^n \ell(h(\boldsymbol{x}_i), y_i). \tag{1}$$

**Neural Network Classifier.** Here, we consider our hypothesis as a neural network and we define it recursively. Given input $\boldsymbol{x} \in \mathcal{X}$, the hidden representations are computed as

$$\boldsymbol{z}_1 = \phi(W^{(1)}\boldsymbol{x}), \tag{2a}$$

$$\boldsymbol{z}_{l+1} = \phi(W^{(l)}\boldsymbol{z}_l) + \sum_{k \in \mathcal{I}_l} \boldsymbol{z}_k, \quad l = 1, \ldots, L-1, \tag{2b}$$

$$h(\boldsymbol{x}) = \text{Softmax}(W^{(L)}\boldsymbol{z}_L), \tag{2c}$$

where $\phi$ is a non-linear activation and $\mathcal{I}_l \subseteq \{1, \ldots, l-1\}$ denotes the set of skip connections into layer $l$. For example, in a 3-layer network, if we add a skip from layer 1 to 3, then $\mathcal{I}_3 = \{1\}$. And since we start from the first layer, $\mathcal{I}_1 = \emptyset$. For analytical tractability, we model skip connections as additive terms after activation. Bias parameters can be concatenated in weights.

**Matrix Normal Distribution.** Skip connections couple the outputs of entire layers, inducing dependencies across full weight matrices. To model such correlations in a tractable way, we adopt the *matrix normal distribution* (MND), which naturally captures row- and column-wise covariance structures.

**Definition 3.1** (Matrix Normal Distribution). Let $X \in \mathbb{R}^{m \times p}$ be a random matrix. Given positive definite covariance matrices $U \in \mathbb{S}_m^{++}$ and $V \in \mathbb{S}_p^{++}$, we say that $X$ follows a matrix normal distribution with mean $M \in \mathbb{R}^{m \times p}$, denoted

$$X \sim \mathcal{MN}_{m,p}(M, U, V),$$

if its density is

$$p(X \mid M, U, V) = \frac{\exp\left(-\frac{1}{2}\text{tr}\left[V^{-1}(X-M)^T U^{-1}(X-M)\right]\right)}{(2\pi)^{mp/2}\det(V)^{m/2}\det(U)^{p/2}}.$$

Equivalently, $\text{vec}(X) \sim \mathcal{N}(\text{vec}(M), V \otimes U)$, where $\otimes$ denotes the Kronecker product, and $\text{vec}(\cdot)$ is the vectorization operation for matrices.

*Remark* 3.2. Matrix-normal priors and posteriors (often with Kronecker-factored covariance) are common in Bayesian deep learning and variational approximations (Ritter et al., 2018; Huang et al., 2020; Schnaus et al., 2023). Here we employ them as a stylized but tractable tool to capture cross-layer dependencies.

**PAC-Bayesian Bound.** The *generalization gap* is the difference between natural and empirical risks (Eq. 1). Although directly computing this gap is infeasible for modern neural networks, PAC-Bayesian theory provides a principled way to bound it in terms of the KL divergence between posterior and prior distributions over weights. We recall McAllester's classical bound (McAllester, 1998; Guedj & Shawe-Taylor, 2019), which forms the basis of our analysis.

**Theorem 3.3** (McAllester's bound). *Given $h \in \mathcal{H}$ and $S = \{(\boldsymbol{x}_i, y_i)\}_{i=1}^n \sim D^n$ be $n$ i.i.d. samples. For any prior distribution $P \in \mathcal{P}(\mathcal{H})$ independent of $S$, and any posterior distribution $Q \in \mathcal{P}(\mathcal{H})$ possibly dependent on $S$, with probability at least $1 - \delta$ over $S$, we have*

$$\forall Q \in \mathcal{P}(\mathcal{H}), \quad \mathbb{E}_{h \sim Q}[R(h)] \leq \mathbb{E}_{h \sim Q}[\widehat{R}(h)] + \sqrt{\frac{KL(Q\|P) + \ln(\frac{\sqrt{n}}{\delta})}{2n}}. \tag{3}$$

This theorem highlights the important role of the KL term in upper-bounding the generalization gap, where it typically serves as a complexity measure. However, existing work rarely captures architectural factors such as skip connections. In the following, we propose a complexity measure that accounts for these factors.

# 4 MAIN RESULTS

In this section, we present the main results of the paper. We begin by recalling the KL divergence for matrix-normal distributions (MNDs), then introduce our proposed measure for cross-layer correlation, followed by two theorems capturing the key correlation patterns observed in MLPs and CNNs.

**Proposition 4.1** (KL divergence between MNDs). *Let $Q = \mathcal{MN}_{m,p}(M_Q, U_Q, V_Q)$ and $P = \mathcal{MN}_{m,p}(M_P, U_P, V_P)$ be two matrix normal distributions with means $M_Q, M_P \in \mathbb{R}^{m \times p}$, row covariances $U_Q, U_P \in \mathbb{S}_m^{++}$, and column covariances $V_Q, V_P \in \mathbb{S}_p^{++}$. Then the KL divergence admits a closed form*

$$\mathrm{KL}(Q\|P) = \frac{1}{2}\mathrm{tr}\big[(V_Q V_P^{-1}) \otimes (U_Q U_P^{-1})\big] + \frac{1}{2}\mathrm{tr}\big[V_P^{-1}(M_Q - M_P)^T U_P^{-1}(M_Q - M_P)\big]$$
$$- \frac{mp}{2} + \frac{m}{2}\log\frac{\det(V_P)}{\det(V_Q)} + \frac{p}{2}\log\frac{\det(U_P)}{\det(U_Q)}. \tag{4}$$

The proof of Prop. 4.1 is deferred to Appendix E.1. Since the weight matrices of neural networks may have different shapes, we first pad them to a common size for notational simplicity. Appendix C shows that padding with non-trainable standard Gaussian entries leaves the KL divergence unchanged. Thus, without loss of generality, we concatenate the weights as $W = (W_1, W_2, \ldots, W_L)$, where for each $l = 1, 2, \ldots, L$, $W_l \in \mathbb{R}^{m \times r_l}$ and $\sum_{l=1}^{L} r_l = p$.

Following standard assumptions in the literature (Jiang et al., 2019; Jin et al., 2020), we take the prior distribution to be $P = \mathcal{MN}_{m,p}(W^{(0)}, \sigma I_m, I_p)$, which corresponds, after vectorization, to an isotropic Gaussian prior $\mathrm{vec}(W) \sim \mathcal{N}(\mathrm{vec}(W^{(0)}), \sigma^2 I_{mp})$. While recent work has explored data-dependent priors for achieving tighter bounds, we adopt this simpler form in order to focus on the effect of skip connections.

**Assumption on posteriors** A full covariance structure for the posterior captures all information contained in the trained neural network. However, estimating such a distribution is typically infeasible in practice, and simplified assumptions are adopted to balance tractability with the ability to capture the most influential factors. Following Jiang et al. (2019); Jin et al. (2020; 2022), we assume that the variance of each parameter is unchanged ($\mathrm{diag}(\Sigma_Q) = \mathrm{diag}(\Sigma_P)$). In contrast to earlier works, we relax two strong assumptions: the isotropy of weight matrices within each layer (Jiang et al., 2019) and the independence of weights across layers (Jin et al., 2020; 2022). Under these settings, the KL divergence simplifies to

$$\mathrm{KL}(Q\|P) = \sum_{l=1}^{L} \mathrm{tr}\left[(W_l^{(F)} - W_l^{(0)})^\top (W_l^{(F)} - W_l^{(0)})\right] + \frac{m}{2}\log\frac{1}{\det(V_Q)} + \frac{p}{2}\log\frac{\sigma^{2m}}{\det(U_Q)}. \tag{5}$$

We follow the notion of weight correlation (Jin et al., 2020) between rows of weight matrix for in a given layer, and extend to correlation across different layers. To simplify the following analysis, we let the size of all weights be the same (i.e., $\forall l, r_l = r$).

## 4.1 GENERAL WEIGHT CORRELATION

We extend the notion of weight correlation Jin et al. (2020) to cover the relation between layers. Therefore, we can analyse its impact on generalization gap.

**Definition 4.2** (General weight correlation). Given weight matrix $W_l, W_s$ at $l$-th and $s$-th layers, the generalized weight correlation is defined as

$$\rho_{l,s} \triangleq \frac{1}{mr}\sum_{i,j=1}^{m} \frac{|W_{l,i}^T W_{s,j}|}{\|W_{l,i}\|_2, \|W_{s,j}\|_2}, \tag{6}$$

where $W_{l,i}$ is the $i$-th row of the matrix $W_l$, corresponding to the $i$-th at $l$-th layer.

We recall the weight correlation (Jin et al., 2020), and show that *weight correlation* is just a special case of our formulation as it measures the same weights. This connection is discussed in Appendix D.

## 4.2 Connection to Flatness of Loss surface

Let $\boldsymbol{\omega} = \mathrm{vec}(W)$ and $\boldsymbol{\omega}^\star$ denote the MAP estimate of the posterior weights. The log-likelihood of the posterior (i.e., $\log p(\boldsymbol{\omega} \mid S)$) can then be approximated by a second-order Taylor expansion, as shown in Eq. 7. This approximation forms the basis for analyzing how skip connections affect posterior correlations and, consequently, generalization.

$$\log p(\boldsymbol{\omega} \mid S) \approx \log p(\boldsymbol{\omega}^* \mid S) - \frac{1}{2}(\boldsymbol{\omega} - \boldsymbol{\omega}^*)^T \mathbb{E}_S[H](\boldsymbol{\omega} - \boldsymbol{\omega}^*) \tag{7}$$

Hence, the posterior can be approximated as Gaussian,

$$\boldsymbol{\omega} = \mathrm{vec}(W) \sim \mathcal{N}(\mathrm{vec}(W^\star), \mathbb{E}_S[H]^{-1}) \tag{8}$$

Computing the inverse of the full Hessian matrix is infeasible. An approximation is to conduct the Kronecker product decomposition, and we have for each weight matrix

$$W_l \sim \mathcal{MN}\left(W_l^*, \mathbb{E}_S[V_{l,l}]^{-1}, \mathbb{E}_S[U]^{-1}\right) \tag{9}$$

## 4.3 Analysis of Cross-Layer Correlation Patterns

To examine the impact of general weight correlation, we further decompose the column-wise correlation $V_Q$ and establish the following lemma. We then investigate two characteristic patterns corresponding to sparse skip connections and dense connections.

**Proposition 4.3.** *Let the weights of neural networks be $W_l \in \mathbb{R}^{m \times r}$, and let matrix $R = (\rho_{i,j})_{i,j}$, defined in Def. 4.2. Let*

$$V_Q = \mathrm{diag}(1 - \rho_{1,1}, \ldots, 1 - \rho_{L,L}) \otimes I + R \otimes J \tag{10}$$

*where $J = \mathbf{1}\mathbf{1}^T$ is the dot product of all one vector $\mathbf{1}$. Then,*

$$\log \det(V_Q) = (r-1) \sum_{l=1}^{L} \log(1 - \rho_{l,l}) + \log \det(\mathrm{diag}(1 - \rho_{1,1}, \ldots, 1 - \rho_{L,L}) + rR). \tag{11}$$

The proof is in Appendix E.2. The weight correlation is just a special case of our formulation by letting $R = \mathrm{diag}(\rho_{1,1}, \rho_{2,2}, \ldots, \rho_{L,L})$. The detailed discussion is in Appendix D.

Def. 4.2 and Prop. 4.3 both assume $r_l = r$ for simplicity. However this assumption can be relaxed with mixed correlation between rows and columns for weights of different layers, allowing mismatch of size for weights.

Now, we consider a case where there is a correlation between adjacent layers. This is particularly the case for MLPs, as is shown in Fig. 2a.

**Proposition 4.4** (Adjacent Connection). *Given the neural network defined in Eq. 2a,2b and 2c and KL divergence in Eq. 20, let $R$ be 1-banded matrix, i.e.,*

$$R = \mathrm{diag}(\rho_{1,1}, \ldots, \rho_{L,L}) + \mathrm{diag}_1(\rho_{1,2}, \ldots, \rho_{L-1,L}) + \mathrm{diag}_{-1}(\rho_{1,2}, \ldots, \rho_{L,L-1}) \tag{12}$$

*where $\mathrm{diag}_1(\cdots)$ and $\mathrm{diag}_{-1}(\cdots)$ are superdiagonal matrices shifted one element from the diagonal. Let*

$$\Delta_L = \det(\mathrm{diag}(1 - \rho_{1,1}, \ldots, 1 - \rho_{L,L}) + rR) \tag{13}$$

*which can be represented recursively as*

$$\Delta_L = [1 + (r-1)\rho_{L,L}]\Delta_{L-1} - r^2 \rho_{L-1,L}^2 \Delta_{L-2} \tag{14}$$

*and for all $l = 2, \ldots, L$,*

$$\frac{\partial \Delta_L}{\partial \rho_{l-1,l}} \leq 0. \tag{15}$$

We provide a more general version of the proposition, with the proof deferred to Appendix E.4. Prop. 4.4 establishes a monotonic relationship between the term $\log \det(V_Q)$ and the correlations $\rho_{l-1,l}$ between adjacent layers, corresponding to the case illustrated in Fig. 2a. For MLPs without skip connections, this relation holds directly; however, introducing a long skip connection can alleviate the effect, as shown in Fig. 2c, resulting in a smaller generalization gap (since it is positively related to the KL divergence).

We also consider the case where correlations across different layers differ only minimally, similar to the scenarios in Fig. 2c, Fig. 2e, and Fig. 2f. Dense connections in 5-layer MLPs (Fig. 4) can be approximated under this setting by using a single scalar to represent all general weight correlations among layers.

**Proposition 4.5** (Homogeneous Connection). *Consider the same conditions in Prop. 4.4, and let*

$$R = \mathrm{diag}(\rho_{1,1}, \dots, \rho_{L,L}) + \rho(J_L - I_L) \tag{16}$$

*where $J_L = \mathbf{1}\mathbf{1}^T$ and $I_L$ is identity matrix of size $L$. Hence, we have*

$$\Delta_L = \prod_{l=1}^{L}(1 + (r-1)\rho_{l,l} - r\rho)\left(1 + \sum_{l=1}^{L}\frac{r\rho}{1 + (r-1)\rho_{l,l} - r\rho}\right) \tag{17}$$

*And for any $l = 1, \dots L$ if*

$$\rho \approx \rho_{l,l} + \frac{1 - \rho_{l,l}}{r} \tag{18}$$

*the derivative of $\Delta_L$ w.r.t $\rho$ will be unstable such that $\Delta'_L(\rho) \to \infty$.*

The proof is provided in Appendix E.4. Prop. 4.5 reveals an interesting phenomenon: as cross-layer correlation approaches a certain point, it can significantly degrade generalization performance. This behavior aligns with the large empirical generalization gaps observed for $\mathrm{MLP}_{2,2,1}(2)$ in Tab. 1 and $\mathrm{CNN}_{3,2,1}(1)$ in Tab. 2.

## 5 EXPERIMENT

| Network | PFN | PSN | PBC | PBGC | $\Delta$ Loss |
|---|---|---|---|---|---|
| $\mathrm{MLP}_{0,0,0}$ | 1.20e+05 | 2.70e+03 | 3.62e+03 | 3.18e+03 | 5.31e-01 ($\pm$7.4e-04) |
| $\mathrm{MLP}_{0,0,1}$ | 1.41e+05 | 4.47e+03 | 3.97e+03 | 3.55e+03 | 4.55e-01 ($\pm$1.3e-04) |
| $\mathrm{MLP}_{0,1,0}(1)$ | 1.31e+05 | 2.86e+03 | 3.74e+03 | 3.22e+03 | 4.75e-01 ($\pm$4.8e-04) |
| $\mathrm{MLP}_{0,1,0}(2)$ | 1.34e+05 | 4.29e+03 | 4.84e+03 | 3.73e+03 | 4.19e-01 ($\pm$3.1e-04) |
| $\mathrm{MLP}_{1,0,0}(1)$ | 1.47e+05 | 4.18e+03 | 3.97e+03 | 3.53e+03 | 4.51e-01 ($\pm$4.2e-04) |
| $\mathrm{MLP}_{1,0,0}(2)$ | 1.36e+05 | 2.51e+03 | 4.39e+03 | 3.73e+03 | 3.67e-01 ($\pm$3.7e-04) |
| $\mathrm{MLP}_{1,0,0}(3)$ | 1.02e+05 | 1.05e+03 | 3.28e+03 | 2.90e+03 | 3.76e-01 ($\pm$9.0e-04) |
| $\mathrm{MLP}_{1,1,1}(1)$ | 7.41e+04 | 3.98e+03 | 5.42e+03 | 8.27e+03 | 4.53e-01 ($\pm$3.9e-03) |
| $\mathrm{MLP}_{2,2,1}(2)$ | 9.64e+06 | 9.37e+06 | 1.38e+04 | 2.20e+04 | 7.32e-02 ($\pm$1.3e-03) |
| $\mathrm{MLP}_{2,2,1}(3)$ | 5.26e+04 | 1.47e+03 | 4.32e+03 | 6.11e+03 | 4.51e-01 ($\pm$1.2e-03) |
| $\mathrm{MLP}_{3,2,1}(1)$ | 6.48e+04 | 9.48e+02 | 4.09e+03 | 5.35e+03 | 4.91e-01 ($\pm$1.2e-03) |
| Kendall $\tau$ | -2.02e-01 | -8.69e-02 | 1.45e-02 | **7.24e-02** | 1 |

Table 1: Selective results for skip connections with different complexity and performance metrics on 5-layer MLPs. This table reports four complexity measures (PFN, PSN, PBC, and PBGC). The full results are provided in Tab. 6. $\Delta$Loss denotes the empirical generalization gap, defined as the difference between test and training loss. Each model is further trained for 5 additional epochs with a small learning rate, and we report the mean and standard deviation across runs. The last row reports Kendall's $\tau$ correlation. Bold numbers indicate the highest value, while underlined numbers correspond to the PBC method.

To study the effect of skip connections on generalization gaps, we trained 5-layer MLPs on Fashion-MNIST and 5-layer CNNs on CIFAR-10 with all possible skip-connection configurations. For CNNs, we consider both versions with and without batch normalization. All MLPs use a hidden

size of 256, while CNNs use 256 channels per layer with $3 \times 3$ kernels. Models are trained for 80 epochs using SGD with a learning rate of $2 \times 10^{-2}$, momentum of 0.9, and weight decay of $10^{-4}$. For the toy example used to compute full covariance matrices, we train a smaller 5-layer MLP with input dimension 8 for 100 epochs. All experiments were run on a single NVIDIA RTX 3090 GPU with Python 3.9.7 and PyTorch 1.9.1.

## 5.1 COMPLEXITY MEASURE

To benchmark our approach, we compare it against several established complexity measures:

- Product of Frobenius Norms (PFN): Defined as the product of Frobenius norms of all weight matrices, PFN reflects the overall magnitude of network parameters across layers.

- Product of Spectral Norms (PSN) Bartlett et al. (2017): Computed as the product of spectral norms of the weight matrices, PSN emphasizes the worst-case layer-wise amplification effect and has been widely studied in generalization bounds.

- PAC-Bayes & Correlation (PBC) Jin et al. (2020): An extension of the PAC-Bayes framework that incorporates weight correlations, capturing richer dependencies among parameters than standard PAC-Bayes bounds.

- We refer to our method as PAC-Bayes & Generalization Correlation (PBGC), which explicitly incorporates the proposed General Weight Correlation (GWC).

For evaluation, we assess the agreement between empirical rankings of generalization performance and those predicted by different complexity measures using Kendall's $\tau$ correlation coefficient (Kendall, 1938). This statistic quantifies rank similarity by comparing the number of concordant and discordant pairs, with values ranging from -1 (complete disagreement) to 1 (perfect agreement).

To present our results clearly, we first introduce the notation for our models. We use MLP and CNN to denote the model type. A superscript $b$, such as $CNN^b$, indicates the use of batch normalization. Skip connections are considered only in hidden layers (as is typical, since classification networks rarely connect hidden layers directly to inputs or output intermediate features). We represent skip-connection patterns with a triple index—for example, $(0, 0, 0)$ denotes the number of connections at each position (corresponding to $|\mathcal{I}_l|$ in Eq. 2b). When multiple connection types share the same number, we use an additional index to distinguish patterns. The detailed notations are summarized in Tab. 5. For cases with a unique configuration, we omit the index for brevity.

## 5.2 RESULTS OF MLP

Tab. 1 summarizes the results for skip connections in 5-layer MLPs. The last row reports Kendall's $\tau$ correlation. As shown, our proposed method achieves the highest Kendall $\tau$ among PFN, PSN, and PBC, indicating that it more effectively captures the influence of skip connections.

Comparing $MLP_{0,0,0}$ with $MLP_{1,0,0}(3)$ in Tab. 1, we observe that $MLP_{1,0,0}(3)$—which includes a long skip connection from the first hidden layer to the last hidden layer—exhibits both a smaller empirical generalization gap (3.76e-01 vs. 5.31e-01) and a lower PBGC measure (2.90e+03 vs. 3.18e+03). Consistently, Fig. 2a and Fig. 2c show that the cross-layer weight correlation is reduced for $MLP_{1,0,0}(3)$. These results provide strong evidence in support of Prop. 4.4. From Fig. 2, it is evident that the general weight correlation effectively reflects the skip connections in MLPs. In contrast, CNNs exhibit markedly different behaviour.

## 5.3 RESULT OF CNN

Unlike MLPs, the impact of skip connections on CNNs is almost negligible. As shown in Fig. 2e and Fig. 2f, the hidden-layer patterns exhibit no discernible differences. Consistently, the generalization gap in Tab. 2 shows only a slight reduction for $CNN_{0,0,0}$ (from 5.31e-01 to 4.53e-01), while both PBC and PBGC increase. This suggests that, for CNNs, skip connections do not primarily act through general weight correlation. Kendall's $\tau$ further supports this observation: although PBGC improves marginally over PBC, it is not the best-performing measure—the highest correlation is

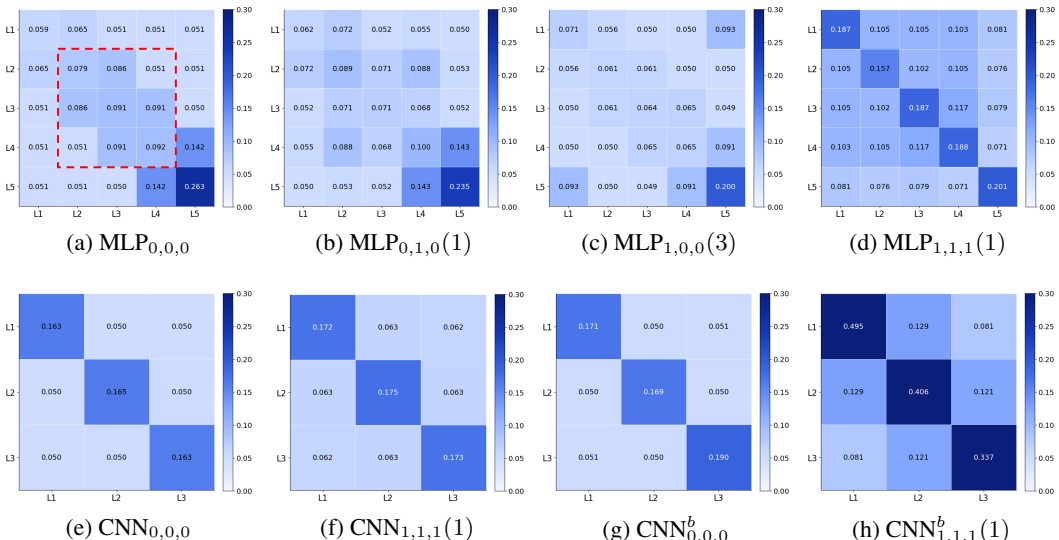

Figure 2: The figures visualize the general weight correlation matrix $R$ defined in Prop. 4.3. For CNNs, the first and last layers are omitted since they are not directly comparable; we therefore compute the general weight correlation only across hidden channels.

achieved by PFN. This implies that the influence of skip connections in CNNs may instead be linked to the norms of the weight matrices.

In contrast, CNNs with batch normalization behave quite differently. As illustrated in Fig. 2g and Fig. 2h, $CNN_{1,1,1}^b$ exhibits patterns similar to $MLP_{0,0,0}$. Interestingly, this indicates that batch-normalized CNNs demonstrate an effect opposite to that observed in MLPs.

We validate the Gaussian approximation using diagnostics in Tab. 3. Parameters are projected onto a 30-dimensional PCA subspace, and we evaluate three standard metrics: the Shapiro–Wilk rejection rate, the Anderson–Darling statistic, and the effective sample size (ESS). As shown in Tab. 1, the Shapiro–Wilk rejection rates are low, the mean ESS values are stable, and the Anderson–Darling statistics are small across all architectures and skip configurations. These results indicate that the Gaussian approximation provides a reliable local model of the posterior in our setting.

We extend our evaluation to ResNet-18 on CIFAR-100 (Tab. 4) and additionally report Spearman and distance correlation (dCor), along with their corresponding p-values. Since ResNet-18 has 256 possible skip-connection configurations, we include several representative examples in the main text and visualize the correlation trends in Fig. 4 (appendix). As shown in Tab. 2, our proposed complexity measure consistently yields the strongest correlations across all three statistics, with significance levels below 1%.

## 6   CONCLUSION AND LIMITATION

We introduced a PAC-Bayesian framework that makes explicit the role of architectural structure in generalization via General Weight Correlation (GWC) and its induced matrix $R$. By Kronecker-factoring the posterior covariance, our method extends weight correlation to capture cross-layer dependencies created by skip connections. The theory shows that adjacent-layer correlations enlarge the KL term and thus hinder generalization, while heterogeneous, layer-specific correlations are beneficial. Empirically, PBGC best aligns (via Kendall's $\tau$) with observed generalization trends across all skip patterns in MLPs, and reveals a contrasting picture for CNNs, where skip connections have limited effect unless batch normalization is present. These results isolate when and how skip connections help from a PAC-Bayesian viewpoint, providing actionable guidance for non-parametric architectural design. The limitation includes extension to more general and complex models, e.g., transformer-based models.

| Network | PFN | PSN | PBC | PBGC | $\Delta$ Loss |
|---------|-----|-----|-----|------|--------|
| $CNN_{0,0,0}$ | 1.20e+05 | 2.70e+03 | 3.62e+03 | 3.18e+03 | 5.31e-01 ($\pm$7.4e-04) |
| $CNN_{0,0,1}$ | 1.41e+05 | 4.47e+03 | 3.97e+03 | 3.55e+03 | 4.55e-01 ($\pm$1.3e-04) |
| $CNN_{0,1,0}(1)$ | 1.31e+05 | 2.86e+03 | 3.74e+03 | 3.22e+03 | 4.75e-01 ($\pm$4.8e-04) |
| $CNN_{0,1,0}(2)$ | 1.34e+05 | 4.29e+03 | 4.84e+03 | 3.73e+03 | 4.19e-01 ($\pm$3.1e-04) |
| $CNN_{1,0,0}(1)$ | 1.47e+05 | 4.18e+03 | 3.97e+03 | 3.53e+03 | 4.51e-01 ($\pm$4.2e-04) |
| $CNN_{1,0,0}(2)$ | 1.36e+05 | 2.51e+03 | 4.39e+03 | 3.73e+03 | 3.67e-01 ($\pm$3.7e-04) |
| $CNN_{1,0,0}(3)$ | 1.02e+05 | 1.05e+03 | 3.28e+03 | 2.90e+03 | 3.76e-01 ($\pm$9.0e-04) |
| $CNN_{1,1,1}(1)$ | 7.41e+04 | 3.98e+03 | 5.42e+03 | 8.27e+03 | 4.53e-01 ($\pm$3.9e-03) |
| $CNN_{2,2,1}(2)$ | 5.32e+04 | 3.19e+04 | 1.13e+04 | 1.32e+05 | 6.90e-01 ($\pm$1.6e-04) |
| $CNN_{2,2,1}(3)$ | 3.76e+04 | 1.72e+04 | 1.00e+04 | 1.03e+05 | 5.70e-01 ($\pm$3.6e-04) |
| $CNN_{3,2,1}(1)$ | 7.85e+04 | 3.19e+04 | 1.07e+04 | 1.11e+05 | 7.90e-01 ($\pm$1.6e-04) |
| Kendall $\tau$ | **2.96e-01** | 2.41e-01 | 2.09e-01 | 2.17e-01* | 1 |

Table 2: Selective results for skip connections with different complexity and performance metrics on 5-layer CNNs. Bold numbers denote the highest values, underlined numbers correspond to the PBC method, and starred numbers indicate our proposed method.

Table 3: **Summary of Distribution Diagnostics (Projected to $k = 30$) Across Skip Configurations.**

| Skip Config | Mean ESS | Min ESS | Shapiro reject rate | Mean AD stat |
|-------------|----------|---------|---------------------|--------------|
| $CNN_{0,0,0}$ | 287.6 | 225.0 | 0.00% | 0.397 |
| $CNN_{1,1,1}(1)$ | 283.8 | 227.1 | 3.33% | 0.436 |
| $CNN_{2,1,1}(1)$ | 270.3 | 163.3 | 0.00% | 0.375 |
| $CNN_{2,2,1}(1)$ | 272.1 | 163.8 | 0.00% | 0.380 |
| $CNN_{3,2,1}(1)$ | 289.1 | 196.0 | 3.33% | 0.352 |
| $CNN^b_{0,0,0}(1)$ | 279.8 | 171.6 | 0.00% | 0.376 |
| $CNN^b_{1,1,1}(1)$ | 300.0 | 300.0 | 0.00% | 0.306 |
| $CNN^b_{2,1,1}(1)$ | 288.3 | 288.2 | 0.00% | 0.442 |
| $CNN^b_{2,2,1}(1)$ | 285.6 | 226.5 | 3.33% | 0.436 |
| $CNN^b_{3,2,1}(1)$ | 281.1 | 219.2 | 0.00% | 0.342 |
| $MLP_{0,0,0}(1)$ | 281.7 | 235.6 | 0.00% | 0.397 |
| $MLP_{1,1,1}(1)$ | 272.9 | 163.0 | 0.00% | 0.368 |
| $MLP_{2,1,1}(1)$ | 270.3 | 158.5 | 3.33% | 0.403 |
| $MLP_{1,2,1}(1)$ | 268.8 | 170.1 | 0.00% | 0.384 |
| $MLP_{2,2,1}(1)$ | 282.3 | 216.7 | 0.00% | 0.347 |
| $MLP_{3,2,1}(1)$ | 270.8 | 105.0 | 0.00% | 0.348 |

Table 4: **Correlation Analysis Between Model Complexity Measures and the Empirical Complexity Gap.** The experiment is conducted on representative skip-connection configurations of ResNet18 on CIFAR-100.

| Measure | Kendall $\tau$ (p-val) | Spearman $\rho$ (p-val) | dCor (p-val) |
|---------|----------------------|------------------------|--------------|
| PFN | $-4.12$e-01 (2e-02) | $-6.37$e-01 (4e-03) | 3.96e-01 (1e-01) |
| PSN | $-4.25$e-01 (1e-02) | $-6.45$e-01 (4e-03) | 4.29e-01 (8e-02) |
| PB | $-4.51$e-01 (9e-03) | $-6.49$e-01 (4e-03) | 5.77e-01 (2e-02) |
| PBC | $4.90$e-01 (4e-03) | $6.66$e-01 (3e-03) | 8.35e-01 (0e+00) |
| **PBGC** | **5.42**e-01 (**1e-03**) | **7.09**e-01 (**9e-04**) | **8.41**e-01 (**0e+00**) |

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

## A  NOTATION OF NEURAL NETWORKS

| Connection Notation | $\mathcal{I}_2$ | $\mathcal{I}_3$ | $\mathcal{I}_4$ |
|---|---|---|---|
| $0,0,0$ | - | - | - |
| $0,0,1$ | - | - | {3} |
| $0,1,0(1)$ | - | {2} | - |
| $0,1,0(2)$ | - | - | {2} |
| $1,0,0(1)$ | {1} | - | - |
| $1,0,0(2)$ | - | {1} | - |
| $1,0,0(3)$ | - | - | {1} |
| $1,1,1(1)$ | {1} | {2} | {3} |
| $1,1,1(2)$ | {1} | - | {2,3} |
| $1,1,1(3)$ | - | {1,2} | {3} |
| $1,1,1(4)$ | - | {1} | {2,3} |
| $1,1,1(5)$ | - | {2} | {1,3} |
| $1,1,1(6)$ | - | - | {1,2,3} |
| $1,2,1(1)$ | {1} | {2} | {2,3} |
| $1,2,1(2)$ | {1,2} | - | {2,3} |
| $1,2,1(3)$ | - | {2} | {1,2,3} |
| $2,1,1(1)$ | {1} | {1,2 } | {3} |
| $2,1,1(2)$ | {1} | {1} | {2,3} |
| $2,1,1(3)$ | {1} | {2} | {1,3} |
| $2,1,1(4)$ | {1} | - | {1,2,3} |
| $2,1,1(5)$ | - | {1,2} | {1,3} |
| $2,1,1(6)$ | - | {1} | {1,2,3} |
| $2,2,1(1)$ | {1} | {1,2} | {2,3} |
| $2,2,1(2)$ | {1} | {2} | {1,2} |
| $2,2,1(3)$ | - | {1,2} | {1,2,3} |
| $3,2,1(1)$ | {1} | {1,2} | {1,2,3} |

Table 5: Notation table for configuration of skip-connections

Here, we provide the notation for the skip-connection configurations corresponding to $\mathcal{I}_l$ in Eq. equation 2b. The use of $\mathcal{I}_l$ is for mathematical rigor. Intuitively, the number in the tuple indicates the starting layer of the skip connection, while the number in parentheses enumerates the different variants.

## B  TOY EXPERIMENT WITH TINY MLPS ON MNIST

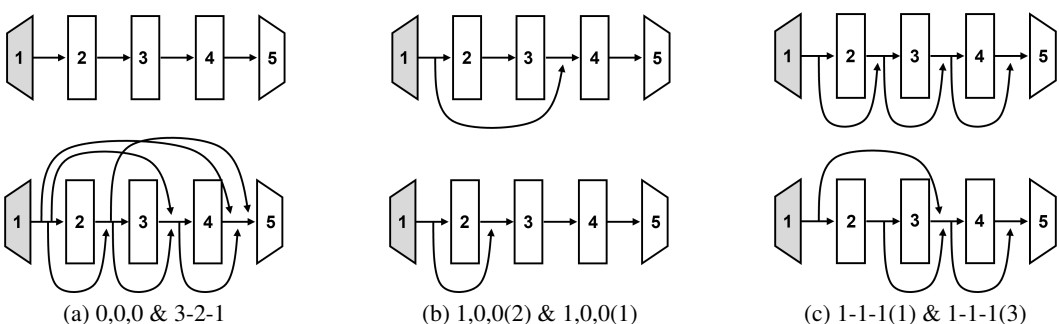

(a) 0,0,0 & 3-2-1          (b) 1,0,0(2) & 1,0,0(1)          (c) 1-1-1(1) & 1-1-1(3)

Figure 3: **Illustration of skip configurations**. The three numbers in the tuple denote the starting points of the skip connections. For example, Fig. 3a shows an MLP without any skip and dense connections. The number in parentheses in Fig. 3b and 3c specifies the detailed skip configuration, which is further explained in Tab. 5.

Here, we provide a toy experiment for 5-Layer MLPs on resized MNIST. The model architectures with different skip-connection configurations are illustrated in Fig. 3. To have enough samples for covariance matrix, we resize the MNIST dataset from $1 \times 28 \times 28$ to $1 \times 4 \times 4$. We set the width of each hidden layer to 8, hence the total number of parameters is controlled around 500. The model is trained for 100 epochs using the Adam optimizer until convergence. After convergence, we continue training with a very small learning rate to induce mild perturbations around the local minimum. During this phase, we record the network parameters after each batch update, resulting in over 2000 samples. To mitigate the drift introduced by the continued optimization, we apply a sliding window of 100 samples to compute the rolling mean and subtract it from the recorded parameters. The empirical covariance matrix is then computed from these de-meaned samples, providing a stable and accurate estimate.

Typical SWAG performs sampling to obtain a low-rank approximation of the posterior covariance. This is crucial for large models, but unnecessary in our setting since the network is small. Therefore, we use the standard unbiased empirical estimator to compute the full covariance matrix. In addition, we employ a sliding-window rolling mean instead of an overall average to better correct for drift during sampling. Fig. 1 presents the resulting covariance matrix after removing the rows and columns corresponding to parameters that do not change around the minimum (such parameters lead to `NaN` entries in the covariance).

Here, we provide the details about how we conduct the estimation of posterior covariance matrix in Fig. 1. We first provide the training details, then show how we estimate the covariance.

## C  PADDING THE WEIGHT MATRIX

We show that padding the weight matrices of a neural network with non-trainable entries does not affect the KL divergence between prior and posterior weight distributions.

Consider an $L$-layer network with weights $\{W_1, W_2, \ldots, W_L\}$ before padding and $\{\widetilde{W}_1, \widetilde{W}_2, \ldots, \widetilde{W}_L\}$ after padding. Let $P$ and $Q$ denote the prior and posterior distributions,

respectively. Define the vectorized parameters

$$\boldsymbol{\omega} = \begin{pmatrix} \text{vec}(W_1) \\ \text{vec}(W_2) \\ \vdots \\ \text{vec}(W_L) \end{pmatrix}, \qquad \widetilde{\boldsymbol{\omega}} = \begin{pmatrix} \text{vec}(\widetilde{W}_1) \\ \text{vec}(\widetilde{W}_2) \\ \vdots \\ \text{vec}(\widetilde{W}_L) \end{pmatrix}, \tag{19}$$

where $\text{vec}(\cdot)$ denotes column-wise vectorization.

The KL divergence between Gaussian posterior $Q = \mathcal{N}(\boldsymbol{\mu}_Q, \Sigma_Q)$ and prior $P = \mathcal{N}(\boldsymbol{\mu}_P, \Sigma_P)$ is

$$\text{KL}(Q\|P) = \frac{1}{2}\left[\log\frac{\det(\Sigma_P)}{\det(\Sigma_Q)} - m + (\boldsymbol{\mu}_Q - \boldsymbol{\mu}_P)^T\Sigma_P^{-1}(\boldsymbol{\mu}_Q - \boldsymbol{\mu}_P) + \text{tr}(\Sigma_P^{-1}\Sigma_Q)\right]. \tag{20}$$

Padding is implemented by augmenting each $W_l, l = 1, 2 \ldots L$ with non-trainable entries (standard Gaussian), so that all weight matrices share the same maximal row/column dimensions. Since padding entries are non-trainable, their quadratic contribution in Eq. 20 cancels, i.e.

$$(\boldsymbol{\mu}_Q - \boldsymbol{\mu}_P)^T\Sigma_P^{-1}(\boldsymbol{\mu}_Q - \boldsymbol{\mu}_P) = (\widetilde{\boldsymbol{\mu}}_Q - \widetilde{\boldsymbol{\mu}}_P)^T\widetilde{\Sigma}_P^{-1}(\widetilde{\boldsymbol{\mu}}_Q - \widetilde{\boldsymbol{\mu}}_P). \tag{21}$$

Let padding be an independent standard Gaussian ($\boldsymbol{\nu} \sim \mathcal{N}(\mathbf{0}, I)$), and re-arrange the variants as

$$\widetilde{\boldsymbol{\omega}} = \begin{pmatrix} \boldsymbol{\omega} \\ \boldsymbol{\nu} \end{pmatrix}. \tag{22}$$

For the covariance structure, this implies

$$\widetilde{\Sigma}_P^{-1}\widetilde{\Sigma}_Q = \begin{pmatrix} \Sigma_P^{-1} & 0 \\ 0 & I \end{pmatrix}\begin{pmatrix} \Sigma_Q & 0 \\ 0 & I \end{pmatrix} = \begin{pmatrix} \Sigma_P^{-1}\Sigma_Q & 0 \\ 0 & I \end{pmatrix}. \tag{23}$$

The determinant factor is likewise preserved:

$$\det(\widetilde{\Sigma}) = \det\begin{pmatrix} \Sigma & 0 \\ 0 & I \end{pmatrix} = \det(\Sigma). \tag{24}$$

Thus all terms in equation 20 remain unchanged under padding. Hence the KL divergence between prior and posterior distributions is invariant to padding.

*Remark* C.1. Padding simply appends additional coordinates that are identically distributed under both the prior and posterior (standard Gaussian, independent of the trainable weights). Since KL divergence only measures discrepancies between two distributions, these extra variables contribute zero to the KL.

## D    CONNECTION TO WEIGHT CORRELATION

We make segmentation for column covariance $V$ according to columns of weights at each layer, and consider the factorization of the covariance matrix for vectorized weights from all layers that $\Sigma = V \otimes U$, we have

$$V \otimes U = \begin{pmatrix} V_{1,1} \otimes U & V_{1,2} \otimes U & \cdots & V_{1,L} \otimes U \\ V_{2,1} \otimes U & V_{2,2} \otimes U & \cdots & V_{1,L} \otimes U \\ \vdots & \vdots & \ddots & \vdots \\ V_{L,1} \otimes U & V_{L,2} \otimes U & \cdots & V_{L,L} \otimes U \end{pmatrix} \tag{25}$$

Let $V_{i,j} = 0, \forall i \neq j$, $U = \sigma^2 I$ and

$$V_{i,i} = \begin{pmatrix} 1 & \rho_i & \cdots & \rho_i \\ \rho_i & 1 & \cdots & \rho_i \\ \vdots & \vdots & \ddots & \vdots \\ \rho_i & \rho_i & \cdots & 1 \end{pmatrix}. \tag{26}$$

We show that $-\log\det(V_{i,i} \otimes U)$ is indeed the weight correlation factor in the KL-divergence. Another notion related to our work is *weight volume* Jin et al. (2022) as defined in D.1.

**Definition D.1** (Weight Volume (Jin et al., 2022)). Let

$$\Sigma_l = \mathbb{E}\left[(\text{vec}(W_l) - \mathbb{E}(\text{vec}(W_l)))(\text{vec}(W_l) - \mathbb{E}(\text{vec}(W_l)))^T\right] \tag{27}$$

be the weight covariance matrix in a neural network. The *weight volume* is defined as

$$\text{vol}(W_l) \triangleq \frac{\det(\Sigma_l)}{\prod_i [\Sigma_l]_{ii}}. \tag{28}$$

This provides a more general notion that accounts for all possible correlations within a given weight matrix. In our setting, it can be estimated as $\text{vol}(W_l) = \det(V_{l,l} \otimes U)$.

# E  OMITTED PROOFS

**Lemma E.1** (KL divergence between MNDs). *Let $Q = \mathcal{MN}_{m,p}(M_Q, U_Q, V_Q)$ and $P = \mathcal{MN}_{m,p}(M_P, U_P, V_P)$ be two matrix normal distributions with means $M_Q, M_P \in \mathbb{R}^{m \times p}$, row covariances $U_Q, U_P \in \mathbb{S}_m^{++}$, and column covariances $V_Q, V_P \in \mathbb{S}_p^{++}$. Then the KL divergence admits a closed form*

$$\text{KL}(Q\|P) = \frac{1}{2}\text{tr}\left[(V_Q V_P^{-1}) \otimes (U_Q U_P^{-1})\right] + \text{tr}\left[V_P^{-1}(M_Q - M_P)^T U_P^{-1}(M_Q - M_P)\right]$$
$$- \frac{mp}{2} + \frac{m}{2}\log\frac{\det(V_P)}{\det(V_Q)} + \frac{p}{2}\log\frac{\det(U_P)}{\det(U_Q)}. \tag{29}$$

*Proof.* Starts from Def. 3.1, we have

$$\text{KL}(Q\|P) = \frac{1}{2}\mathbb{E}_Q \text{tr}\left[V_P^{-1}(X - M_P)^T U_P^{-1}(X - M_P) - V_Q^{-1}(X - M_Q)^T U_Q^{-1}(X - M_Q)\right] \tag{30}$$

$$+ \frac{m}{2}\log\frac{\det(V_P)}{\det(V_Q)} + \frac{p}{2}\log\frac{\det(U_P)}{\det(U_Q)} \tag{31}$$

$$= \frac{1}{2}\mathbb{E}_Q \text{tr}\left[V_P^{-1}(X - M_Q + M_Q - M_P)^T U_P^{-1}(X - M_Q + M_Q - M_P)\right] \tag{32}$$

$$- \frac{1}{2}\mathbb{E}_Q\left[\text{vec}(X - M_Q)^T(V_Q^{-1} \otimes U_Q^{-1})\text{vec}(X - M_Q)\right] \tag{33}$$

$$+ \frac{m}{2}\log\frac{\det(V_P)}{\det(V_Q)} + \frac{p}{2}\log\frac{\det(U_P)}{\det(U_Q)} \tag{34}$$

$$= \frac{1}{2}\mathbb{E}_Q \text{tr}\left[V_P^{-1}(X - M_Q)^T U_P^{-1}(X - M_Q)\right] + \frac{1}{2}\text{tr}\left[V_P^{-1}(M_Q - M_P)^T U_P^{-1}(M_Q - M_P)\right] \tag{35}$$

$$- \frac{mp}{2} + \frac{m}{2}\log\frac{\det(V_P)}{\det(V_Q)} + \frac{p}{2}\log\frac{\det(U_P)}{\det(U_Q)} \tag{36}$$

$$= \frac{1}{2}\text{tr}[(V_Q V_P^{-1}) \otimes (U_Q U_P^{-1})] + \frac{1}{2}\text{tr}\left[V_P^{-1}(M_Q - M_P)^T U_P^{-1}(M_Q - M_P)\right] \tag{37}$$

$$- \frac{mp}{2} + \frac{m}{2}\log\frac{\det(V_P)}{\det(V_Q)} + \frac{p}{2}\log\frac{\det(U_P)}{\det(U_Q)} \tag{38}$$

$$\square$$

**Lemma E.2.** *Let $A, B \in \mathbb{R}^{L \times L}$ and $J \in \mathbb{S}_r$. Then,*

$$\det\left(A \otimes I_r + B \otimes J\right) = \prod_{i=1}^{r} \det\left(A + \lambda_i B\right). \tag{39}$$

*where $I_r$ is the identity matrix of size $r$.*

*Proof.* Let $Q$ be the orthogonal matrix diagonalizing $J$, i.e., $Q^T J Q = \mathrm{diag}(\lambda_1, \ldots, \lambda_r) = \Lambda$. By similarity invariance of the determinant,

$$\det\left(A \otimes I_r + B \otimes J\right) = \det\left((I_L \otimes Q)^T (A \otimes I_r + B \otimes J)(I_L \otimes Q)\right). \tag{40}$$

Using the mixed-product property of Kronecker products, this equals

$$\det\left(A \otimes I_r + B \otimes \Lambda\right). \tag{41}$$

Consider *commutation matrix* $K$ such that

$$\det\left(A \otimes I_r + B \otimes \Lambda\right) = \det\left(K(A \otimes I_r + B \otimes \Lambda)K^T\right) \tag{42}$$

$$= \det\left(I_r \otimes A + \Lambda \otimes B\right) \tag{43}$$

Hence the determinant factorizes as

$$\prod_{i=1}^{r} \det(A + \lambda_i B). \tag{44}$$

$\square$

**Lemma E.3** (Determinant of block correlation matrix with heterogeneous sizes). *Let $r_1, \ldots, r_L \in \mathbb{N}$ and define*

$$V = \mathrm{diag}\left((1 - \rho_{1,1})I_{r_1}, \ldots, (1 - \rho_{L,L})I_{r_L}\right) + \left(\rho_{l,k}\, J_{r_l, r_k}\right)_{l,k=1}^{L}, \tag{45}$$

*where $J_{r_l, r_k} = \mathbf{1}_{r_l}\mathbf{1}_{r_k}^T$. Let*

$$D = \mathrm{diag}(1 - \rho_{1,1}, \ldots, 1 - \rho_{L,L}), \qquad R = (\rho_{l,k}\sqrt{r_l r_k})_{l,k=1}^{L}. \tag{46}$$

*Hence,*

$$\log \det(V) = \sum_{l=1}^{L}(r_l - 1)\log(1 - \rho_{l,l}) + \log \det\left(D + R\right). \tag{47}$$

*Proof.* For each block $l$, define $u_l = \mathbf{1}_{r_l}/\sqrt{r_l}$ and extend it to an orthogonal basis $Q_l = [u_l U_l] \in \mathbb{R}^{r_l \times r_l}$. Then,

$$Q_l^T I_{r_l} Q_l = I_{r_l}, \qquad Q_l^T J_{r_l, r_k} Q_k = \begin{pmatrix} \sqrt{r_l r_k} & \cdots & 0 \\ \vdots & \ddots & \vdots \\ 0 & \cdots & 0 \end{pmatrix}. \tag{48}$$

Let $Q = \mathrm{diag}(Q_1, \ldots, Q_L)$. By similarity invariance of the determinant, for the second term in Eq. 45. we have

$$\mathrm{diag}(Q_1^T, \cdots, Q_L^T)\left(\rho_{l,k} J_{r_l, r_k}\right)_{l,k=1}^{L}\mathrm{diag}(Q_1, \cdots, Q_L) = \tag{49}$$

$$\begin{pmatrix} Q_1^T & 0 & \cdots & 0 \\ 0 & Q_2^T & \cdots & 0 \\ \vdots & \vdots & \ddots & \vdots \\ 0 & 0 & \cdots & Q_L^T \end{pmatrix} \begin{pmatrix} \rho_{1,1}J_{r_1,r_1} & \rho_{1,2}J_{r_1,r_2} & \cdots & \rho_{1,L}J_{r_1,r_L} \\ \rho_{2,1}J_{r_2,r_1} & \rho_{2,2}J_{r_2,r_2} & \cdots & \rho_{2,L}J_{r_2,r_L} \\ \vdots & \vdots & \ddots & \vdots \\ \rho_{L,1}J_{r_L,r_1} & \rho_{L,2}J_{r_L,r_2} & \cdots & \rho_{L,L}J_{r_L,r_L} \end{pmatrix} \begin{pmatrix} Q_1 & 0 & \cdots & 0 \\ 0 & Q_2 & \cdots & 0 \\ \vdots & \vdots & \ddots & \vdots \\ 0 & 0 & \cdots & Q_L \end{pmatrix}$$

$$\tag{50}$$

$$= \begin{pmatrix} \rho_{1,1}r_1\mathbf{e}_{1,1} & \rho_{1,2}\sqrt{r_1 r_2}\mathbf{e}_{1,2} & \cdots & \rho_{1,L}\sqrt{r_1 r_L}\mathbf{e}_{1,L} \\ \rho_{2,1}\sqrt{r_2 r_1}\mathbf{e}_{2,1} & \rho_{2,2}r_2\mathbf{e}_{2,2} & \cdots & \rho_{2,L}\sqrt{r_2 r_L}\mathbf{e}_{1,L} \\ \vdots & \vdots & \ddots & \vdots \\ \rho_{L,1}\sqrt{r_L r_1}\mathbf{e}_{L,1} & \rho_{L,2}\sqrt{r_L r_2}\mathbf{e}_{L,1} & \cdots & \rho_{L,L}r_L\mathbf{e}_{L,L} \end{pmatrix} \tag{51}$$

where $\mathbf{e}_{l,k} \in \mathbb{R}^{l \times k}$ denotes the matrix with first elements of 1 and others are all 0. Hence, with a commutative matrix $K$, such that

$$\det(V) = \det(KVK^T) = \left(\prod_{\ell=1}^{L}(1 - \rho_{l,l})^{r_l - 1}\right)\det(D + R). \tag{52}$$

$\square$

Here, we recall Prop. 4.5 and provides the proof.

**Proposition E.4.** *Consider the same conditions in Prop. 4.4, and let*

$$R = \text{diag}(\rho_{1,1}, \ldots, \rho_{L,L}) + \rho(J_L - I_L) \tag{53}$$

*where $J_L = \mathbf{1}\mathbf{1}^T$ and $I_L$ is identity matrix of size L. Hence, we have*

$$\Delta_L = \prod_{l=1}^{L}(1 + (r-1)\rho_{l,l} - r\rho)\left(1 + \sum_{l=1}^{L}\frac{r\rho}{1 + (r-1)\rho_{l,l} - r\rho}\right) \tag{54}$$

*And for any $l = 1, \ldots L$ if*

$$\rho \approx \rho_{l,l} + \frac{1 - \rho_{l,l}}{r} \tag{55}$$

*the derivative of $\Delta_L$ w.r.t $\rho$ will be unstable such that $\Delta_L'(\rho) \to \infty$.*

*Proof.* Since

$$\Delta_L = \det\left(\text{diag}(1 - \rho_{1,1}, \ldots, 1 - \rho_{L,L}) + rR\right) \tag{56}$$

$$= \det\left(\text{diag}(1 + (r-1)\rho_{1,1} - r\rho, \ldots, 1 + (r-1)\rho_{L,L} - r\rho) + r\rho J_L\right) \tag{57}$$

$$= \det\left(\text{diag}(1 + (r-1)\rho_{1,1} - r\rho, \ldots, 1 + (r-1)\rho_{L,L} - r\rho) + r\rho\mathbf{1}\mathbf{1}^T\right) \tag{58}$$

$$= \prod_{l=1}^{L}(1 + (r-1)\rho_{l,l} - r\rho)\left(1 + r\rho\mathbf{1}^T\Lambda^{-1}\mathbf{1}\right) \tag{59}$$

where $\Lambda = \text{diag}(1 - r\rho + (r-1)\rho_{1,1}, \ldots, 1 - r\rho + (r-1)\rho_{L,L})$. Hence,

$$\Delta_L = \prod_{l=1}^{L}(1 + (r-1)\rho_{l,l} - r\rho)\left(1 + \sum_{l=1}^{L}\frac{r\rho}{1 + (r-1)\rho_{l,l} - r\rho}\right) \tag{60}$$

Now, we show the derivative of $\Delta$ w.r.t $\rho$. Let us consider $\widetilde{\rho} = r\rho$

$$A(\widetilde{\rho}) = \sum_{l=1}^{L}\frac{1}{1 + (r-1)\rho_{l,l} - \widetilde{\rho}} \tag{61}$$

we have

$$A'(\widetilde{\rho}) = \sum_{l=1}^{L}\frac{1}{(1 + (r-1)\rho_{l,l} - \widetilde{\rho})^2} \tag{62}$$

Then take logarithm on $\Delta_L$ and take derivative

$$\frac{\Delta_L'(\widetilde{\rho})}{\Delta_L(\widetilde{\rho})} = \sum_{l=1}^{L}\frac{-1}{(1 + (r-1)\rho_{l,l} - \widetilde{\rho})} + \frac{A(\widetilde{\rho}) + \widetilde{\rho}A'(\widetilde{\rho})}{1 + \widetilde{\rho}A(\widetilde{\rho})} \tag{63}$$

$$= -A(\widetilde{\rho}) + \frac{A(\widetilde{\rho}) + \rho A'(\widetilde{\rho})}{1 + \widetilde{\rho}A(\widetilde{\rho})} \tag{64}$$

$$= \frac{\widetilde{\rho}(A'(\widetilde{\rho}) - A^2(\widetilde{\rho}))}{1 + \widetilde{\rho}A(\widetilde{\rho})} \tag{65}$$

Since $\widetilde{\rho} = r\rho$,

$$\Delta_L'(\rho) = \Delta_L\frac{r^2\rho(A'(r\rho) - A^2(r\rho))}{1 + r\rho A(r\rho)} \tag{66}$$

The sign of the derivative depends on

$$A'(r\rho) - A^2(r\rho) = -\sum_{l\neq s}\frac{1}{(1 + (r-1)\rho_{l,l} - r\rho)(1 + (r-1)\rho_{s,s} - r\rho)} \tag{67}$$

Notice that $\rho = \frac{1 + (r-1)\rho_{l,l}}{r}$ should be avoid or it will be instable. $\qquad\square$

### E.1 PROOF OF PROP. 4.4

In this section, to accommodate a more general prior distribution, we establish a broader proposition in place of Prop. 4.4, from which Lem. 4.4 follows as a direct consequence.

**Proposition E.5.** *Let $\boldsymbol{\omega}_l = vec(W_l) \in \mathbb{R}^{N_l N_{l-1}}, l \in [L]$ be the vectorized weight matrix on $l$-th layer, $P$ be fixed prior Gaussian probability measure and $Q$ be the posterior Gaussian probability that dependent of the training process. We assume that the covariance matrices for $P$ and $Q$ are*

$$\Sigma_P = \begin{pmatrix} \sigma_{P,1}^2 I & 0 & 0 & \cdots & 0 \\ 0 & \sigma_{P,2}^2 I & 0 & \cdots & 0 \\ 0 & 0 & \sigma_{P,3}^2 I & \cdots & 0 \\ \vdots & \vdots & \vdots & \ddots & \vdots \\ 0 & 0 & 0 & \cdots & \sigma_{P,L}^2 I \end{pmatrix}, \Sigma_Q = \begin{pmatrix} \sigma_{Q,1}^2 I & K_{1,2} & 0 & \cdots & 0 \\ K_{1,2}^T & \sigma_{Q,2}^2 I & K_{2,3} & \cdots & 0 \\ 0 & K_{2,3}^T & \sigma_{Q,3}^2 I & \cdots & 0 \\ \vdots & \vdots & \vdots & \ddots & \vdots \\ 0 & 0 & 0 & \cdots & \sigma_{Q,L}^2 I \end{pmatrix} \tag{68}$$

*where $\sigma_{P,l}^2 I, \sigma_{Q,l}^2 I$ are covariance matrices of $\boldsymbol{\omega}_l$ on probability measure $P$ and $Q$ separately. $K_{l,s}, l, s \in [L]$ denotes the cross-covariance. Assume that $\Sigma_Q$ is not degenerated. We further assume that each pair of elements between adjacent layers shares the same correlation coefficient, we have*

$$K_{l-1,l} = \sigma_{\rho,l-1} \sigma_{\rho,l} \rho_{l-1,l} \mathbf{1}_{N_{l-1}, N_l} \tag{69}$$

*where $\mathbf{1}_{N_{l-1}, N_l}$ is $N_{l-1} \times N_l$ matrix each element of which is $1$. Therefore, we have*

$$KL(\rho\|\pi) = \frac{1}{2} \sum_{l=1}^{L} \left( \frac{\|\mathbb{E}_\rho[\boldsymbol{\omega}_l] - \mathbb{E}_\pi[\boldsymbol{\omega}_l]\|_2^2}{\sigma_{\pi,l}^2} + N_l N_{l-1} \left( \frac{\sigma_{\rho,l}^2}{\sigma_{\pi,l}^2} + \log \frac{\sigma_{\pi,l}^2}{\sigma_{\rho,l}^2} - 1 \right) \right) - \log \prod_{l=1}^{L} \det(A_l) \tag{70}$$

$$\tag{71}$$

*and $\det(A_l)$ is determined by the recursive difference equation*

$$\det(A_l) = 1 - \frac{N_{l-1} N_l \tau_{l-1,l}^2}{\det(A_{l-1})} \tag{72}$$

*and we have $\frac{\partial KL(\rho\|\pi)}{\partial \tau_{l-1,l}^2} \geq 0$ showing that the KL-divergence will increase as each $\rho_{l-1,l}^2$ increases.*

*Proof.* Assume that $\Sigma_Q$ is not degenerated, and let $\boldsymbol{\omega}$ be the concatenation of all vectorized weight matrices and $\mu_P = \mathbb{E}_P[\boldsymbol{\omega}], \mu_Q = \mathbb{E}_Q[\boldsymbol{\omega}]$ for simplicity. Hence, the KL divergence for $Q$ and $P$ is

$$KL(Q\|P) = \frac{1}{2} \mathbb{E}_Q \left[ \log \frac{\det(\Sigma_P)}{\det(\Sigma_Q)} - (\boldsymbol{\omega} - \mu_Q)^T \Sigma_Q^{-1} (\boldsymbol{\omega} - \mu_Q) + (\boldsymbol{\omega} - \mu_P)^T \Sigma_P^{-1} (\boldsymbol{\omega} - \mu_P) \right] \tag{73}$$

$$= \frac{1}{2} \left[ \log \frac{\det(\Sigma_P)}{\det(\Sigma_Q)} - \sum_{l=1}^{L} N_l N_{l-1} + (\mu_Q - \mu_P)^T \Sigma_P^{-1} (\mu_Q - \mu_P) + \text{tr} \left( \Sigma_P^{-1} \Sigma_Q \right) \right] \tag{74}$$

$$= \frac{1}{2} \left[ \log \frac{\det(\Sigma_P)}{\det(\Sigma_Q)} - \sum_{l=1}^{L} N_l N_{l-1} + \sum_{l=1}^{L} \frac{\|\mathbb{E}_Q[\boldsymbol{\omega}_l] - \mathbb{E}_P[\boldsymbol{\omega}_l]\|_2^2}{\sigma_{P,l}^2} + \text{tr} \left( \Sigma_P^{-1} \Sigma_Q \right) \right] \tag{75}$$

$$= \frac{1}{2} \left[ \log \frac{\det(\Sigma_P)}{\det(\Sigma_Q)} + \sum_{l=1}^{L} \left( \frac{\|\mathbb{E}_Q[\boldsymbol{\omega}_l] - \mathbb{E}_P[\boldsymbol{\omega}_l]\|_2^2}{\sigma_{P,l}^2} + N_l N_{l-1} \left( \frac{\sigma_{Q,l}^2}{\sigma_{P,l}^2} - 1 \right) \right) \right] \tag{76}$$

where $N_0$ denotes the input dimension. In order to approximate $\log \det(\Sigma_Q)$, we try to triangularize $\Sigma_Q$, and we have

$$
\det\left(\Sigma_Q\right) = \det
\begin{pmatrix}
I & 0 & \cdots & 0 \\
-\frac{K_{1,2}^T}{\sigma_{Q,1}^2} & I & \cdots & 0 \\
\vdots & \vdots & \ddots & \vdots \\
0 & 0 & \cdots & I
\end{pmatrix}
\begin{pmatrix}
\sigma_{Q,1}^2 I & K_{1,2} & \cdots & 0 \\
K_{1,2}^T & \sigma_{Q,2}^2 I & \cdots & 0 \\
\vdots & \vdots & \ddots & \vdots \\
0 & 0 & \cdots & \sigma_{Q,L}^2 I
\end{pmatrix}
\tag{77}
$$

$$
= \det
\begin{pmatrix}
\sigma_{Q,1}^2 I & K_{1,2} & \cdots & 0 \\
0 & \sigma_{Q,2}^2 I - \frac{K_{1,2}^T K_{1,2}}{\sigma_{Q,1}^2} & \cdots & 0 \\
\vdots & \vdots & \ddots & \vdots \\
0 & 0 & \cdots & \sigma_{Q,L}^2 I
\end{pmatrix}.
\tag{78}
$$

Let $A_1 = I$ and $A_2 = I - \frac{K_{1,2}^T K_{1,2}}{\sigma_{Q,1}^2 \sigma_{Q,2}^2}$ and it is invertible, since we assume that all the eigenvalues in $\frac{K_{1,2}^T K_{1,2}}{\sigma_{Q,1}^2 \sigma_{Q,2}^2}$ are between $[0,1)$. This is quite reasonable. If violated, some weights could be entirely represented by others. we have

$$
\det\left(\Sigma_Q\right) = \sigma_{Q,1}^{2N_1 N_0} \det
\begin{pmatrix}
\sigma_{Q,2}^2 A_2 & K_{2,3} & \cdots & 0 \\
K_{2,3}^T & \sigma_{Q,3}^2 I & \cdots & 0 \\
\vdots & \vdots & \ddots & \vdots \\
0 & 0 & \cdots & \sigma_{Q,L}^2 I
\end{pmatrix}
\tag{79}
$$

$$
= \sigma_{Q,1}^{2N_1 N_0} \det
\begin{pmatrix}
I & 0 & \cdots & 0 \\
-\frac{K_{2,3}^T}{\sigma_{Q,2}^2} A_2^{-1} & I & \cdots & 0 \\
\vdots & \vdots & \ddots & \vdots \\
0 & 0 & \cdots & I
\end{pmatrix}
\begin{pmatrix}
\sigma_{Q,2}^2 A_2 & K_{2,3} & \cdots & 0 \\
K_{2,3}^T & \sigma_{Q,3}^2 I & \cdots & 0 \\
\vdots & \vdots & \ddots & \vdots \\
0 & 0 & \cdots & \sigma_{Q,L}^2 I
\end{pmatrix}
\tag{80}
$$

$$
= \sigma_{Q,1}^{2N_1 N_0} \det
\begin{pmatrix}
\sigma_{Q,2}^2 A_2 & K_{2,3} & \cdots & 0 \\
0 & \sigma_{Q,3}^2 I - \frac{K_{2,3}^T A_2^{-1} K_{2,3}}{\sigma_{Q,2}^2} & \cdots & 0 \\
\vdots & \vdots & \ddots & \vdots \\
0 & 0 & \cdots & \sigma_{Q,L}^2 I
\end{pmatrix}
\tag{81}
$$

Define $A_l = I - \frac{K_{l-1,l}^T A_{l-1}^{-1} K_{l-1,l}}{\sigma_{Q,l-1}^2 \sigma_{Q,l}^2}, l \in [L]$ and continue doing this we have

$$
\det\left(\Sigma_Q\right) = \prod_{l=1}^{L} \sigma_{Q,l}^{2N_l N_{l-1}} \det(A_l).
\tag{82}
$$

Since, $A_1 = I$ and let $\widetilde{\rho}_{l-1,l}^2 = N_{l-1} N_l \rho_{l-1,l}^2$ for simplicity, we have for $l = 2$

$$
A_2 = I - \rho_{1,2}^2 \mathbf{1}_{N_2,N_1}^T \mathbf{1}_{N_1,N_2}
\tag{83}
$$

$$
= I - N_1 N_2 \rho_{1,2}^2 \frac{1}{N_2} \mathbf{1}_{N_2,N_2}
\tag{84}
$$

$$
= I - \widetilde{\rho}_{1,2}^2 \frac{1}{N_2} \mathbf{1}_{N_2,N_2}
\tag{85}
$$

by the *Neuman series* and the fact $\det(A_2) = 1 - \widetilde{\rho}_{1,2}^2$, we have

$$
A_2^{-1} = \sum_{n=0}^{\infty} \left(\widetilde{\rho}_{1,2}^2\right)^n \frac{1}{N_2} \mathbf{1}_{N_2,N_2}
\tag{86}
$$

$$
= \frac{1}{1 - \widetilde{\rho}_{1,2}^2} \frac{1}{N_2} \mathbf{1}_{N_2,N_2}
\tag{87}
$$

$$
= \frac{1}{\det(A_2)} \frac{1}{N_2} \mathbf{1}_{N_2,N_2}
\tag{88}
$$

and also

$$\det(A_2) = 1 - \frac{\widetilde{\rho}_{1,2}^2}{\det(A_1)}. \tag{89}$$

By induction let

$$A_{l-1}^{-1} = \frac{1}{\det(A_{l-1})} \frac{1}{N_{l-1}} \mathbf{1}_{N_{l-1},N_{l-1}} \tag{90}$$

Hence,

$$A_l = I - \rho_{l-1,l}^2 \mathbf{1}_{N_l,N_{l-1}}^T A_{l-1}^{-1} \mathbf{1}_{N_{l-1},N_l} \tag{91}$$

$$= I - \frac{\widetilde{\rho}_{l-1,l}^2}{\det(A_{l-1})} \frac{1}{N_l} \mathbf{1}_{N_l,N_l} \tag{92}$$

and

$$\det(A_l) = 1 - \frac{\widetilde{\rho}_{l-1,l}^2}{\det(A_{l-1})} = 1 - \frac{N_{l-1}N_l \rho_{l-1,l}^2}{\det(A_{l-1})} \tag{93}$$

Now we prove that $\frac{\partial KL(\rho\|\pi)}{\partial \rho_{l-1,l}^2} \geq 0$. To this end, we only need to prove that $\frac{\partial \prod_{l=1}^L \det(A_l)}{\partial \rho_{l-1,l}^2} \leq 0$. Since $\det(A_l)$ recursively depends on all $\rho_{s-1,s}^2$ by $\det(A_s), s < l$. Hence by *China rule*

$$\frac{\partial \prod_{l=1}^L \det(A_l)}{\partial \rho_{s-1,s}^2} = \prod_{l=1}^{s-1} \det(A_l) \frac{\partial \prod_{l=s}^L \det(A_l)}{\partial \rho_{s-1,s}^2} \tag{94}$$

$$= \prod_{l=1}^{s-1} \det(A_l) \left( \prod_{l=s+1}^L \det(A_l) + \frac{\widetilde{\rho}_{s,s+1}^2}{\det(A_s)} \prod_{l=s+2}^L \det(A_l) + \cdots + \prod_{l=s}^{L-1} \frac{\widetilde{\rho}_{l,l+1}^2}{\det(A_l)} \right) \frac{\partial \det(A_s)}{\partial \rho_{s-1,s}^2} \tag{95}$$

and because $A_l \succ 0, l \in [L]$ is positive definite, we have $\det(A_l) > 0$. Hence, the sign of the above equation depends on

$$\frac{\partial \det(A_s)}{\partial \rho_{s-1,s}^2} = -\frac{N_{s-1}N_s}{\det(A_{s-1})} < 0 \tag{96}$$

**Discussion on** $A_l \succ 0$ Here we explain why $A_l \succ 0$. We start from $A_2$. According to Eq. equation 83, we claim that $\widetilde{\rho}_{1,2}^2 < 1$ which represent the total variance of weights at first layer that can be explained by the second layer. We assume that none of the weights at the first layer can be totally explained by the second layer. $\qquad\square$

# F  ADDITIONAL EXPERIMENTS

In this section, we add some additional experiments to support our conclusion. The Tab. 6 and 6 are the complete experiment results regarding 5-layer MLPs and CNNs separately. Fig. 4 shows the heatmaps for MLPs and CNNs with dense skip configurations.

Tab. 8 summarizes the results of different skip-connection configurations for ResNet-18. The network contains eight skip connections in total, each represented by a binary indicator of 0 or 1. We remove selected skip connections (denoted by 0) and compute several complexity measures—including an additional spectrum-based metric (PSN)—and report their Kendall's $\tau$, Spearman's $\rho$, and dCor correlations with the empirical generalization gap. As shown, our proposed measure achieves the highest correlation across all three metrics, indicating that it effectively captures the inter-layer interactions.

Tab. 9 presents the extended experiments across different architectures. As shown, our measure continues to achieve the highest Kendall's $\tau$.

Fig. 5 and Fig. 6 visualize the complexity measures. The red dashed line denotes the empirical generalization gap.

Table 6: **Comparison of Skip-Connection Configurations in 5-Layer MLPs on Fashion MNIST.** We omit some of the configurations, since they cannot achieve comparable performance. All models are trained with similar accuracy, and the Kendall method is provided to see whether our method indeed captures the influence of skip-connection.

| Network | PFN | PSN | #Param | PBC | PBGC | WC | GWC | Loss | Acc. | Δ Loss |
|---|---|---|---|---|---|---|---|---|---|---|
| $MLP_{0,0,0}(1)$ | 1.20e+05 | 2.70e+03 | 4.00e+05 | 3.62e+03 | 3.18e+03 | 3.15e+03 | 2.71e+03 | 5.40e-01 | 8.98e+01 | 5.31e-01 (±7.4e-04) |
| $MLP_{0,0,1}(1)$ | 1.41e+05 | 4.47e+03 | 4.00e+05 | 3.97e+03 | 3.55e+03 | 3.49e+03 | 3.07e+03 | 4.80e-01 | 8.90e+01 | 4.55e-01 (±1.3e-04) |
| $MLP_{0,1,0}(1)$ | 1.31e+05 | 2.86e+03 | 4.00e+05 | 3.74e+03 | 3.22e+03 | 3.26e+03 | 2.74e+03 | 4.90e-01 | 8.95e+01 | 4.75e-01 (±4.8e-04) |
| $MLP_{0,1,0}(2)$ | 1.34e+05 | 4.29e+03 | 4.00e+05 | 4.84e+03 | 3.73e+03 | 4.35e+03 | 3.25e+03 | 4.50e-01 | 8.97e+01 | 4.19e-01 (±3.1e-04) |
| $MLP_{1,0,0}(1)$ | 1.47e+05 | 4.18e+03 | 4.00e+05 | 3.97e+03 | 3.53e+03 | 3.48e+03 | 3.04e+03 | 4.70e-01 | 8.96e+01 | 4.51e-01 (±4.2e-04) |
| $MLP_{1,0,0}(2)$ | 1.36e+05 | 2.51e+03 | 4.00e+05 | 4.39e+03 | 3.73e+03 | 3.90e+03 | 3.24e+03 | 4.10e-01 | 9.00e+01 | 3.67e-01 (±3.7e-04) |
| $MLP_{1,0,0}(3)$ | 1.02e+05 | 1.05e+03 | 4.00e+05 | 3.28e+03 | 2.90e+03 | 2.83e+03 | 2.45e+03 | 4.10e-01 | 8.93e+01 | 3.76e-01 (±9.0e-04) |
| $MLP_{1,1,1}(1)$ | 7.41e+04 | 3.98e+03 | 4.00e+05 | 5.42e+03 | 8.27e+03 | 4.74e+03 | 7.59e+03 | 4.90e-01 | 8.94e+01 | 4.53e-01 (±3.9e-03) |
| $MLP_{1,1,1}(2)$ | 4.80e+04 | 2.72e+03 | 4.00e+05 | 4.96e+03 | 7.31e+03 | 4.28e+03 | 6.63e+03 | 5.20e-01 | 8.88e+01 | 4.82e-01 (±3.1e-03) |
| $MLP_{1,1,1}(3)$ | 4.77e+04 | 2.51e+03 | 4.00e+05 | 4.90e+03 | 7.11e+03 | 4.22e+03 | 6.44e+03 | 5.20e-01 | 8.86e+01 | 4.76e-01 (±2.8e-03) |
| $MLP_{1,1,1}(4)$ | 5.57e+04 | 2.99e+03 | 4.00e+05 | 4.80e+03 | 5.86e+03 | 4.11e+03 | 5.17e+03 | 5.30e-01 | 8.90e+01 | 4.92e-01 (±3.8e-03) |
| $MLP_{1,1,1}(5)$ | 3.20e+04 | 1.89e+03 | 4.00e+05 | 4.90e+03 | 5.74e+03 | 4.22e+03 | 5.05e+03 | 5.30e-01 | 8.91e+01 | 4.97e-01 (±3.7e-03) |
| $MLP_{1,1,1}(6)$ | 1.88e+04 | 9.34e+02 | 4.00e+05 | 4.60e+03 | 5.56e+03 | 3.93e+03 | 4.89e+03 | 4.90e-01 | 8.92e+01 | 4.48e-01 (±3.5e-03) |
| $MLP_{1,2,1}(1)$ | 6.87e+04 | 3.12e+03 | 4.00e+05 | 4.52e+03 | 6.61e+03 | 3.92e+03 | 6.01e+03 | 5.10e-01 | 8.92e+01 | 4.62e-01 (±8.0e-04) |
| $MLP_{1,2,1}(2)$ | 1.15e+05 | 2.24e+03 | 4.00e+05 | 3.99e+03 | 3.92e+03 | 3.34e+03 | 3.26e+03 | 6.20e-01 | 8.92e+01 | 6.14e-01 (±7.8e-04) |
| $MLP_{1,2,1}(3)$ | 7.81e+04 | 2.03e+03 | 4.00e+05 | 4.01e+03 | 3.91e+03 | 3.36e+03 | 3.26e+03 | 6.80e-01 | 8.90e+01 | 6.75e-01 (±1.2e-03) |
| $MLP_{2,1,1}(1)$ | 7.15e+04 | 2.56e+03 | 4.00e+05 | 4.66e+03 | 6.63e+03 | 4.06e+03 | 6.03e+03 | 5.10e-01 | 8.92e+01 | 4.61e-01 (±1.5e-03) |
| $MLP_{2,1,1}(2)$ | 7.37e+04 | 2.33e+03 | 4.00e+05 | 4.19e+03 | 5.05e+03 | 3.58e+03 | 4.44e+03 | 5.10e-01 | 8.92e+01 | 4.73e-01 (±1.4e-03) |
| $MLP_{2,1,1}(3)$ | 6.54e+04 | 2.25e+03 | 4.00e+05 | 4.38e+03 | 5.88e+03 | 3.78e+03 | 5.28e+03 | 5.10e-01 | 8.90e+01 | 4.59e-01 (±2.8e-03) |
| $MLP_{2,1,1}(4)$ | 5.97e+04 | 1.98e+03 | 4.00e+05 | 4.28e+03 | 4.80e+03 | 3.67e+03 | 4.19e+03 | 5.60e-01 | 8.88e+01 | 5.16e-01 (±1.9e-03) |
| $MLP_{2,1,1}(5)$ | 5.66e+04 | 1.92e+03 | 4.00e+05 | 4.37e+03 | 5.98e+03 | 3.77e+03 | 5.39e+03 | 5.30e-01 | 8.88e+01 | 4.79e-01 (±2.8e-03) |
| $MLP_{2,1,1}(6)$ | 6.02e+04 | 1.99e+03 | 4.00e+05 | 4.35e+03 | 6.16e+03 | 3.75e+03 | 5.56e+03 | 5.20e-01 | 8.84e+01 | 4.63e-01 (±2.1e-03) |
| $MLP_{2,2,1}(2)$ | 9.64e+06 | 9.37e+06 | 4.00e+05 | 1.38e+04 | 2.20e+04 | 1.14e+04 | 1.97e+04 | 4.50e-01 | 8.43e+01 | 7.32e-02 (±1.3e-03) |
| $MLP_{2,2,1}(3)$ | 5.26e+04 | 1.47e+03 | 4.00e+05 | 4.32e+03 | 6.11e+03 | 3.73e+03 | 5.52e+03 | 5.00e-01 | 8.88e+01 | 4.51e-01 (±1.2e-03) |
| $MLP_{3,2,1}(1)$ | 6.48e+04 | 9.48e+02 | 4.00e+05 | 4.09e+03 | 5.35e+03 | 3.57e+03 | 4.83e+03 | 5.50e-01 | 8.84e+01 | 4.91e-01 (±1.2e-03) |
| Kendall | -2.02e-01 | -8.69e-02 | nan | 1.45e-02 | **7.24e-02** | -4.34e-02 | **7.25e-02** | nan | nan | nan |

Table 7: **Comparison of skip connection configurations CNNs on CIFAR10.**

| Network | PFN | PSN | #Param | PBC | PBV | WC | GWC | Loss | Acc. | Δ Loss |
|---|---|---|---|---|---|---|---|---|---|---|
| $CNN_{0,0,0}(1)$ | 2.20e+04 | 6.50e+03 | 4.40e+06 | 9.47e+03 | 8.95e+04 | 9.12e+03 | 8.91e+04 | 1.00e+00 | 6.58e+01 | 3.70e-01 (±2.5e-04) |
| $CNN_{0,0,1}(1)$ | 2.42e+04 | 7.20e+03 | 4.40e+06 | 9.47e+03 | 8.94e+04 | 9.11e+03 | 8.90e+04 | 9.97e-01 | 6.67e+01 | 5.10e-01 (±1.3e-04) |
| $CNN_{0,1,0}(1)$ | 2.47e+04 | 7.30e+03 | 4.40e+06 | 9.45e+03 | 8.90e+04 | 9.10e+03 | 8.87e+04 | 1.05e+00 | 6.59e+01 | 6.40e-01 (±1.7e-04) |
| $CNN_{0,1,0}(2)$ | 2.70e+04 | 8.00e+03 | 4.40e+06 | 9.49e+03 | 9.02e+04 | 9.14e+03 | 8.98e+04 | 1.04e+00 | 6.57e+01 | 5.40e-01 (±1.4e-04) |
| $CNN_{1,0,0}(2)$ | 2.72e+04 | 8.10e+03 | 4.40e+06 | 9.49e+03 | 9.02e+04 | 9.14e+03 | 8.98e+04 | 1.10e+00 | 6.42e+01 | 6.40e-01 (±2.3e-04) |
| $CNN_{1,0,0}(3)$ | 2.92e+04 | 9.40e+03 | 4.40e+06 | 9.54e+03 | 9.21e+04 | 9.19e+03 | 9.17e+04 | 1.10e+00 | 6.15e+01 | 4.60e-01 (±1.5e-04) |
| $CNN_{1,1,1}(1)$ | 3.47e+04 | 1.70e+04 | 4.40e+06 | 1.02e+04 | 1.07e+05 | 9.84e+03 | 1.07e+05 | 1.07e+00 | 6.23e+01 | 3.90e-01 (±1.6e-04) |
| $CNN_{1,1,1}(2)$ | 3.38e+04 | 1.55e+04 | 4.40e+06 | 9.84e+03 | 9.86e+04 | 9.48e+03 | 9.82e+04 | 1.07e+00 | 6.19e+01 | 4.00e-01 (±7.5e-05) |
| $CNN_{1,1,1}(3)$ | 3.06e+04 | 1.04e+04 | 4.40e+06 | 9.61e+03 | 9.32e+04 | 9.26e+03 | 9.29e+04 | 1.08e+00 | 6.23e+01 | 4.40e-01 (±1.5e-04) |
| $CNN_{1,1,1}(4)$ | 3.29e+04 | 1.47e+04 | 4.40e+06 | 9.57e+03 | 9.18e+04 | 9.22e+03 | 9.14e+04 | 1.06e+00 | 6.26e+01 | 3.90e-01 (±1.4e-04) |
| $CNN_{1,1,1}(5)$ | 3.13e+04 | 1.14e+04 | 4.40e+06 | 9.68e+03 | 9.52e+04 | 9.33e+03 | 9.48e+04 | 1.07e+00 | 6.31e+01 | 4.40e-01 (±6.0e-05) |
| $CNN_{1,1,1}(6)$ | 3.04e+04 | 1.03e+04 | 4.40e+06 | 9.53e+03 | 9.15e+04 | 9.18e+03 | 9.11e+04 | 1.08e+00 | 6.22e+01 | 4.30e-01 (±1.1e-04) |
| $CNN_{1,2,1}(1)$ | 4.43e+04 | 2.74e+04 | 4.40e+06 | 1.18e+04 | 1.44e+05 | 1.14e+04 | 1.44e+05 | 1.03e+00 | 6.49e+01 | 4.90e-01 (±2.2e-04) |
| $CNN_{1,2,1}(3)$ | 3.21e+04 | 1.36e+04 | 4.40e+06 | 9.71e+03 | 9.48e+04 | 9.35e+03 | 9.44e+04 | 1.06e+00 | 6.28e+01 | 4.20e-01 (±1.3e-04) |
| $CNN_{2,1,1}(1)$ | 4.26e+04 | 2.11e+04 | 4.40e+06 | 1.05e+04 | 1.12e+05 | 1.01e+04 | 1.11e+05 | 1.02e+00 | 6.57e+01 | 5.40e-01 (±1.9e-04) |
| $CNN_{2,1,1}(2)$ | 4.08e+04 | 2.29e+04 | 4.40e+06 | 1.01e+04 | 1.05e+05 | 9.77e+03 | 1.05e+05 | 1.02e+00 | 6.55e+01 | 5.30e-01 (±1.4e-04) |
| $CNN_{2,1,1}(3)$ | 3.79e+04 | 1.81e+04 | 4.40e+06 | 1.00e+04 | 1.03e+05 | 9.68e+03 | 1.02e+05 | 1.05e+00 | 6.43e+01 | 5.60e-01 (±3.2e-04) |
| $CNN_{2,1,1}(4)$ | 3.75e+04 | 2.02e+04 | 4.40e+06 | 1.02e+04 | 1.07e+05 | 9.81e+03 | 1.06e+05 | 1.06e+00 | 6.38e+01 | 5.70e-01 (±1.3e-04) |
| $CNN_{2,1,1}(5)$ | 3.82e+04 | 1.46e+04 | 4.40e+06 | 9.82e+03 | 9.76e+04 | 9.46e+03 | 9.72e+04 | 1.06e+00 | 6.43e+01 | 6.10e-01 (±1.3e-04) |
| $CNN_{2,1,1}(6)$ | 3.72e+04 | 1.33e+04 | 4.40e+06 | 9.77e+03 | 9.69e+04 | 9.41e+03 | 9.65e+04 | 1.04e+00 | 6.49e+01 | 5.90e-01 (±1.7e-04) |
| $CNN_{2,2,1}(2)$ | 5.32e+04 | 3.19e+04 | 4.40e+06 | 1.13e+04 | 1.32e+05 | 1.09e+04 | 1.32e+05 | 1.09e+00 | 6.43e+01 | 6.90e-01 (±1.6e-04) |
| $CNN_{2,2,1}(3)$ | 3.76e+04 | 1.72e+04 | 4.40e+06 | 1.00e+04 | 1.03e+05 | 9.69e+03 | 1.03e+05 | 1.03e+00 | 6.51e+01 | 5.70e-01 (±3.6e-04) |
| $CNN_{3,2,1}(1)$ | 7.85e+04 | 3.19e+04 | 4.40e+06 | 1.07e+04 | 1.11e+05 | 1.03e+04 | 1.11e+05 | 1.13e+00 | 6.45e+01 | 7.90e-01 (±1.6e-04) |
| Kendall | 2.96e-01 | 2.41e-01 | nan | 2.09e-01 | **2.17e-01** | 2.10e-01 | **2.18e-01** | nan | nan | nan |

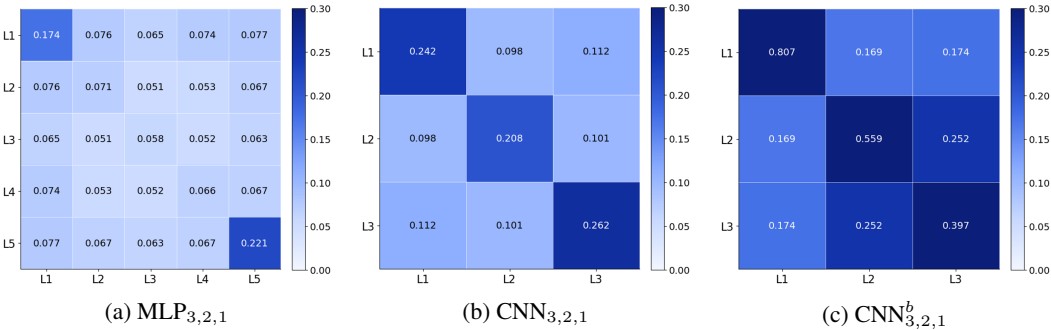

(a) $MLP_{3,2,1}$     (b) $CNN_{3,2,1}$     (c) $CNN^b_{3,2,1}$

Figure 4: The visualization of general weight correlation $R$ for dense connections. We show the dense connections on 5-Layer MLPs, CNNs and CNNs with batch norms.

Table 8: **Correlation Between Complexity Measures and the Empirical Complexity Gap.** We select several representative skip-connection configurations for ResNet-18. The network contains eight shortcuts, where "1" indicates the presence of a shortcut and "0" denotes its removal. In addition to Kendall's $\tau$, we also include Spearman's $\rho$ and distance-based correlation measures.

| Skip-Config. | PFN | PSN | PB | PBC | PBGC | Empirical Gap |
|---|---|---|---|---|---|---|
| 00-00-00-00 | 1.08e+20 | 5.01e+19 | 5.17e+03 | 1.05e+05 | 1.15e+05 | -3.61e+01 |
| 00-00-00-10 | 8.33e+18 | 3.64e+18 | 4.91e+03 | 1.04e+05 | 1.12e+05 | -3.64e+01 |
| 00-00-11-11 | 1.85e+19 | 5.92e+18 | 4.69e+03 | 1.12e+05 | 1.30e+05 | -4.36e+01 |
| 00-11-11-00 | 3.76e+18 | 1.34e+18 | 4.92e+03 | 1.18e+05 | 1.42e+05 | -4.24e+01 |
| 00-11-11-11 | 8.08e+17 | 2.53e+17 | 4.53e+03 | 1.12e+05 | 1.31e+05 | -4.26e+01 |
| 01-01-01-01 | 2.41e+19 | 8.17e+18 | 5.07e+03 | 1.15e+05 | 1.36e+05 | -4.21e+01 |
| 01-11-11-11 | 2.77e+17 | 8.64e+16 | 4.49e+03 | 1.12e+05 | 1.30e+05 | -4.19e+01 |
| 10-00-00-00 | 1.13e+20 | 5.07e+19 | 5.18e+03 | 1.06e+05 | 1.15e+05 | -3.69e+01 |
| 10-01-10-01 | 4.43e+20 | 1.56e+20 | 5.09e+03 | 1.12e+05 | 1.30e+05 | -4.17e+01 |
| 10-10-10-10 | 6.32e+19 | 2.41e+19 | 4.81e+03 | 1.13e+05 | 1.31e+05 | -4.12e+01 |
| 11-00-00-11 | 1.03e+20 | 4.01e+19 | 4.97e+03 | 1.08e+05 | 1.20e+05 | -4.13e+01 |
| 11-00-11-11 | 1.89e+18 | 6.28e+17 | 4.61e+03 | 1.13e+05 | 1.32e+05 | -4.29e+01 |
| 11-01-10-11 | 1.86e+19 | 6.38e+18 | 4.73e+03 | 1.10e+05 | 1.25e+05 | -4.19e+01 |
| 11-11-00-00 | 4.32e+20 | 1.71e+20 | 5.21e+03 | 1.10e+05 | 1.24e+05 | -4.11e+01 |
| 11-11-00-11 | 3.23e+19 | 1.09e+19 | 4.84e+03 | 1.09e+05 | 1.23e+05 | -4.11e+01 |
| 11-11-11-00 | 1.07e+18 | 3.76e+17 | 4.84e+03 | 1.17e+05 | 1.41e+05 | -4.18e+01 |
| 11-11-11-10 | 1.81e+17 | 6.26e+16 | 4.53e+03 | 1.16e+05 | 1.38e+05 | -4.24e+01 |
| 11-11-11-11 | 1.40e+17 | 4.49e+16 | 4.47e+03 | 1.12e+05 | 1.31e+05 | -4.20e+01 |
| Kendall $\tau$ | -4.12e-01 | -4.25e-01 | -4.51e-01 | 4.90e-01 | **5.42e-01** | 1 |
| Spearman $\rho$ | -6.37e-01 | -6.45e-01 | -6.49e-01 | 6.66e-01 | **7.09e-01** | 1 |
| dCor | 3.96e-01 | 4.29e-01 | 5.77e-01 | 8.35e-01 | **8.41e-01** | 1 |

Table 9: **Correlation Between Complexity Measures and the Empirical Complexity Gap Across Architectures.** As shown, our proposed measure consistently achieves the highest Kendall's $\tau$ across different architectures. In addition, we also include the wall clock run time in the table.

| DL Models | PFN | PSN | # Params | PB | PBC | PBGC | Emprical Gap | Run Time(s) |
|---|---|---|---|---|---|---|---|---|
| ResNet50 | 3.96e+58 | 1.44e+58 | 2.37e+07 | 2.75e+04 | 1.84e+06 | 1.84e+05 | 2.96e-01 | 12.4 |
| DenseNet121 | 2.40e+75 | 4.42e+75 | 6.97e+06 | 1.17e+04 | 3.09e+05 | 2.82e+05 | 3.20e-01 | 9.62 |
| VGG16 | 2.65e+19 | 5.42e+17 | 1.35e+08 | 1.24e+04 | 1.04e+05 | 2.12e+04 | 2.80e-01 | 1.77 |
| WRN50 | 2.08e+63 | 6.78e+62 | 6.70e+07 | 3.78e+04 | 3.26e+06 | 2.33e+05 | 2.91e-01 | 21.81 |
| Kendell's $\tau$ | **6.67e-01** | **6.67e-01** | -1 | -3.33e-01 | 0 | **6.67e-01** | 1 | |

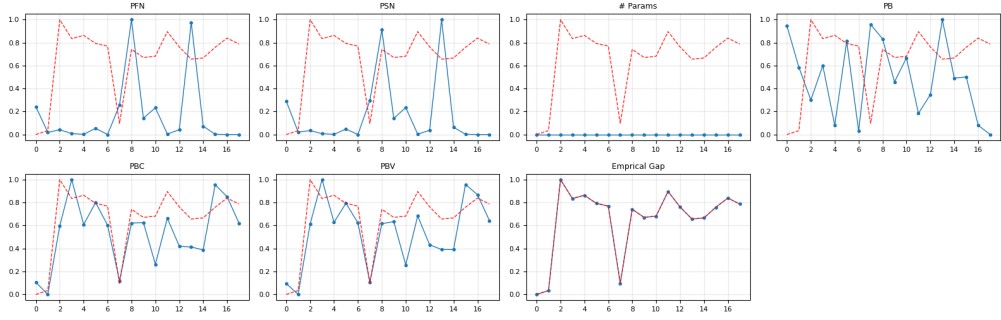

Figure 5: **Comparison of Complexity Measures Across Skip-Connection Configurations in ResNet-18 on CIFAR-100.** We normalize each measure to the range $[0, 1]$ to enable better comparison.

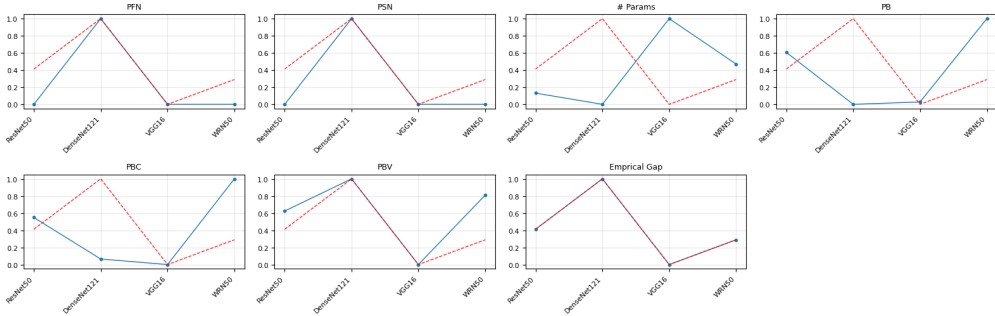

Figure 6: **Comparison of Complexity Measures Across Architectures on CIFAR-100.** We normalize each measure to the range $[0, 1]$ to facilitate better comparison.

