# OpenReview forum: "Skip Connections and Generalization: A PAC-Bayesian Perspective"
_ICLR.cc/2026/Conference — Submitted to ICLR 2026_

### Official Review · Reviewer_SfTR · 2025-10-27

**Soundness:** 2
**Presentation:** 2
**Contribution:** 2
**Rating:** 4
**Confidence:** 2

**Summary:**

The authors relate inter-layer correlations of posterior weight matrices to the KL term in the PAC-Bayes generalization bound by modeling the posterior weight distribution as a Kronecker-factored matrix normal distribution. These correlations are related to the presence or absence of skip connections in the neural network, as evidenced by some example plots. The authors then validate their approach by showing that it more closely tracks the empirical generalization gap than other complexity measures when the presence of skip connections between layers is varied.

**Strengths:**

- As far as I am aware, this is the first attempt to relate inter-layer weight correlations to PAC-Bayes generalization bounds. (However, I am not very familiar with the PAC-Bayes literature, and so not confident in this statement.)
- The theory seems sound, though I did not check the derivations carefully.
- For MLPs, experiment shows that the authors' proposed complexity measure tracks the generalization gap better than previous approaches.

**Weaknesses:**

- The paper is framed as showing a relationship between skip connections and generalization, via posterior weight correlations. However, as far as I can tell, there is no theoretical justification and fairly little empirical justification for the claim that the presence of skip connections relates to weight correlations in a well-understood way.
- The experimental results for CNNs do not favor the approach proposed in this paper.
- Presentation could be improved; in particular Section 4 feels somewhat disorganized and hard to follow.
- The empirics section (5) lacks important details (see Questions).

**Questions:**

- 235: "A full covariance structure for the posterior captures all information contained in the trained neural network". This is not literally true, right? The posterior is not generally Gaussian, so there may be higher-moment information not captured by just the covariance; you are choosing to model it as Gaussian.
	- I'm not sure how justified it is to model the posterior as Gaussian; neural networks are singular so Bernstein-von Mises does not apply[1]. Is Gaussianity a standard assumption in the PAC-Bayes literature?
- It should be explicitly stated at the start of 4 what $U$, $V$, and $V_{ll}$ are. I'm guessing that they are the parameters coming from fitting a Kronecker-factored matrix normal to the posterior distribution of the stacked weights $W$?
	- In particular, it should be emphasized that $V$ encodes the correlation structure of the posterior of $W$.
- 278:  I don't understand $\text{vol}(W_l)=V_{l,l}\otimes U$; the LHS should be a scalar but the RHS looks like a matrix.
- 284: What is $\mathbb{E}_S[H]$? Expected Hessian of $p(\omega\mid S)$?
- What is the precise definition of the "PBGC" number used in Section 5's experiments? How is it computed?
- I don't understand how you go from inter-layer weight correlations to a PAC-Bayes bound, since 4.5 and 4.6 are only show monotonicity instead of anything quantitative. Do you fit $V_Q$ and compute its log det? How?
- What are the relevance of Definitions 4.2 and 4.3? I don't see general weight correlation or weight volume used later in the section. The matrix $V_Q$ used later is not the same as either of these, right?
	- In particular, my understanding is that $V_Q$ captures *posterior* covariance between *columns* of $W$, while general weight correlation is a measure of *row* cosine similarity *given a fixed $W$*
- Overall, Section 4 is kind of hard to follow. I recommend:
	- More handholding: explicitly point out that the point of (13) is to provide an expression for the 2nd term in (5), and that (16) is the exp of the part of (13) that depends on $\rho_{l-1,l}$. Thus Prop 4.5 and 4.6 show that one term of the PAC-Bayes bound (3) is decreasing in $\rho_{l-1,l}$.
	- If they are not directly necessary to the main argument, connections to weight correlation and weight volume should be moved to a separate, later, section, possibly in the appendix.
- Is there a known theoretical relationship between skip connections and inter-layer weight correlations?
	- Figs 1 and 2 hint at some empirical relationship between skip connections and correlations, but it's hard to figure out precisely what's going on by just eyeballing a few plots. I would recommend some more careful empirical studies here.
	- E.g., if you use adjacent-layer skips, how close is the resulting correlation matrix to the 1-banded structure assumed in Prop 4.5?
- How were the posterior correlations in Fig 1 estimated? Did you use SGLD? [2]
- Tables 1 and 2 would be much more readable as line plots with generalization gap on the x-axis and complexity measure on the y axis.

[1] Watanabe "Algebraic Geometry and Statistical Learning Theory." 2011.

[2] Welling et al. "Bayesian Learning via Stochastic Gradient Langevin Dynamics". 2011.

Nit:
- 116: "2rd" -> "2nd"
- 162: "Neural Networks" -> "Neural Network" (twice)
- 172: "starts" -> "start"
- 238: "is identical" -> "is unchanged by training" ?
- 245: $W$ -> $M$ ?
- 254 "between layer" -> "between layers"
- 256 "Given weight matrices $W_l$, $W_s$ at the $l$-th and $s$-th layers"
- 265: Is the second sentence in this paragraph just saying the same thing as the first?
- 269: "relates" -> "related"
- 275: "$\ell$" -> "$l$"
- 295: "Let the weights of the neural network be [...]"
- 299: "Thus" -> "Then"

---

> ### Author Response · Authors · 2025-11-26
> **Response to the Reviewers**
>
> We thank the reviewer for the detailed assessment of our work and for the thoughtful comments and suggestions provided. We greatly appreciate the time and care taken in evaluating the manuscript.
>
> ## How Skip Connections Shape Correlation Structure and Generalization
> Below we clarify how our theoretical results and empirical findings together support the relationship between skip connections, cross-layer correlations, and generalization.
>
> - Lemma 4.4 introduces a closed-form PAC-Bayesian complexity measure that is sensitive to non-parametric architectural factors such as skip connections. This measure provides the analytical foundation for quantifying cross-layer dependencies induced by different architectural choices. As demonstrated in Tab. 1 and 2, it consistently captures the generalization behavior of both MLPs and CNNs across a range of skip-connection patterns.
>
> - To further illustrate these dependencies, Fig. 2 shows that different skip-connection patterns give rise to distinct correlation structures. For example, MLPs without skip connections (Fig. 2a) and batch-normalized CNNs with adjacent skip connections (Fig. 2h) exhibit a clear super-diagonal pattern, reflecting strong correlations between adjacent layers. Prop. 4.5 formalizes this observation by characterizing such banded structures and showing that stronger adjacent-layer correlations are associated with a larger generalization gap. Moreover, Figs. 2b–2c demonstrate that introducing even a single skip connection can alleviate this effect. This trend is consistent with Tab. 2, where architectures up to MLP$_{1,1,1}$ exhibit reduced generalization gaps.
>
> - Conversely, Prop. 4.6 shows that overly dense skip connectivity can induce excessive cross-layer coupling, which in turn can impair generalization. This effect is also evident in Tab. 1 and 2, where densely skip-connected MLPs and CNNs exhibit noticeably worse generalization performance. For example, CNN$_{3,2,1}(1)$ achieves a generalization gap of 0.79 (see Fig. 3c in the Appendix), illustrating the detrimental impact of over-connected skip patterns.
>
> Taken together, these theoretical results and empirical observations provide a coherent explanation of how skip-connection patterns shape inter-layer correlation structures and how these structures relate to generalization. We will revise the presentation of the propositions and accompanying empirical observations to make the arguments clearer and easier to follow in the revised manuscript.
>
> ## Additional Experiments on CNNs
> The results for CNNs stem from the fact that simple CNNs without batch normalization are much less influenced by skip connections, as shown in Fig. 2(e–f). Moreover, for small models on CIFAR10, different skip configurations tend to yield similar performance. To provide a more representative evaluation, we therefore conducted additional experiments on ResNet18 with various skip-connection patterns, as reported below.
>
> (i) We extend our evaluation to ResNet-18 on CIFAR-100 (Tab. 1) and additionally report Spearman and distance correlation (dCor) with their p-values. Because ResNet-18 has 256 possible skip-connection configurations, we include several representative ones in the main text and visualize the correlation trends in Fig. 4 (appendix). As shown in Tab. 2, our proposed complexity measure achieves the strongest correlations across all three statistics, with significance levels below 1%.
>
> **Table 1: Correlation Analysis Between Model Complexity Measures and the Empirical Complexity Gap**
> *The experiment is conducted on representative skip-connection configurations of ResNet18 on CIFAR-100.*
> |**Measure**|**Kendallτ(p-val)**|**Spearmanρ(p-val)**|**dCor(p-val)**|
> |-|-|-|-|
> |PFN|-4.12e-01(2e-02)|-6.37e-01(4e-03)|3.96e-01(1e-01)|
> |PSN|-4.25e-01(1e-02)|-6.45e-01(4e-03)|4.29e-01(8e-02)|
> |PB|-4.51e-01(9e-03)|-6.49e-01(4e-03)|5.77e-01(2e-02)|
> |PBC|4.90e-01(4e-03)|6.66e-01(3e-03)|8.35e-01(0e+00)|
> |**PBGC**|**5.42e-01(1e-03)**|**7.09e-01(9e-04)**|**8.41e-01(0e+00)**|
>
> (ii) We extend our evaluation to multiple architectures. The results are shown in Tab. 2 and Fig. 5(appendix). As illustrated, our complexity measure most consistently tracks the accuracy-gap variation across architectures and achieves the highest Kendall’s
> $\tau$. The complete table is shown in Tab. 7 in appendix.
>
> **Table 2: Extended experiments on CIFAR100**
> |**DLModels**|**PB**|**PBC**|**PBGC**|
> |-|-|-|-|
> |ResNet50|2.75e+04|1.84e+06|1.84e+05|
> |DenseNet121|1.17e+04|3.09e+05|2.82e+05|
> |VGG16|1.24e+04|1.04e+05|2.12e+04|
> |WRN50|3.78e+04|3.26e+06|2.33e+05|
> |**Kendall'sτ**|-3.33e-01|0|**6.67e-01**|
>
> ## Presentation in Section 4
> We thank the reviewer for this helpful comment. We acknowledge that the presentation of Section 4 can be improved. In the revised version, we will reorganize the section to clarify the motivation behind each definition and result, streamline the logical flow, and move some technical details to the appendix to enhance readability.

---

> ### Author Response · Authors · 2025-11-26
> **Response to the Reviewers (Continue)**
>
> ## About the Questions
> - **Q1**. (Line235) We thank the reviewer for raising this important point. We agree that the posterior of a neural network is not globally Gaussian. In PAC-Bayesian analyses, Gaussian posteriors are commonly employed precisely because they yield a tractable closed-form KL term, and such approximations are typically interpreted as locally accurate rather than globally exact.
>
> To justify this assumption in our setting, we provide diagnostic checks in Tab. 2. These results indicate that, in the neighborhood of the trained solution, the posterior landscape is well approximated by a Gaussian distribution, which supports the use of a matrix-normal Laplace approximation for our complexity measure. We will revise the manuscript to clarify this modeling assumption and its scope.
>
> **Table 2: Summary of Diagnostics (Projected to \(k = 30\)) Across Skip Configurations**
> |**SkipConfig**|**MeanESS**|**ShapiroRejectRate**|**MeanADStat**|
> |-|-|-|-|
> |CNN$_{0,0,0}$|287.6|0.00%|0.397|
> |CNN$_{1,1,1}(1)$|283.8|3.33%|0.436|
> |CNN$_{2,1,1}(1)$|270.3|0.00%|0.375|
> |CNN$_{2,2,1}(1)$|272.1|0.00%|0.380|
> |CNN$_{3,2,1}(1)$|289.1|3.33%|0.352|
> |CNN$^b_{0,0,0}(1)$|279.8|0.00%|0.376|
> |CNN$^b_{1,1,1}(1)$|300.0|0.00%|0.306|
> |CNN$^b_{2,1,1}(1)$|288.3|0.00%|0.442|
> |CNN$^b_{2,2,1}(1)$|285.6|3.33%|0.436|
> |CNN$^b_{3,2,1}(1)$|281.1|0.00%|0.342|
> |MLP$_{0,0,0}(1)$|281.7|0.00%|0.397|
> |MLP$_{1,1,1}(1)$|272.9|0.00%|0.368|
> |MLP$_{2,1,1}(1)$|270.3|3.33%|0.403|
> |MLP$_{1,2,1}(1)$|268.8|0.00%|0.384|
> |MLP$_{2,2,1}(1)$|282.3|0.00%|0.347|
> |MLP$_{3,2,1}(1)$|270.8|0.00%|0.348|
>
> - **Q2**.  We agree that the notation in the beginning of Section 4 should be made explicit. In the revised version, we will clearly state the $U$, $V$. We will revise Section 4 to make this interpretation clear from the outset.
>
> - **Q3** (Line 278) This is a typographical error. the correct expression is  $\mathrm{vol}(W_l) = \det(V_{l,l} \otimes U)$.
>
> - **Q4** (Line284) Yes, it refers to the expected Hessian. We will clarify this more clearly in the revised version.
>
> - **Q5** It is computed directly as the second term in Eq. 5, following the method used by Jin et al. [1]. We will include an explicit expression for clarity.
>
> - **Q6** What we compute is a complexity measure derived from the KL divergence in Theorem 3.3. The matrix $V_Q$ is obtained from Lemma 4.4, using the general weight correlation defined in Definition 4.2. Our computation follows the methodology introduced by Jin et al. [1]. We will clarify this with adequte detial in our revised version.
>
> - **Q7** There is no direct relation between Definitions 4.2 and 4.3, as the latter appears in the section *“Connection to Flatness of the Loss Surface.”* Definition 4.3 is included to establish the link between our proposed idea and the formulation in Eq. 11. Our approach is motivated by the Kronecker product decomposition of the posterior weight matrix.
>
> - **Q8** We thank the reviewer for these helpful suggestions. In the revision, we will clarify the role of Eq. (13) as the explicit form of the second term in Eq. (5), explain that Eq. (14) is the part depending on $\rho_{l-1,l}$, and highlight how Props. 4.5 and 4.6 describe its variation in the PAC-Bayes bound. We will also reorganize nonessential material to improve readability. These revisions will make the structure of Section 4 clearer.
>
> - **Q9** To our knowledge, no existing theory directly links specific skip-connection patterns to inter-layer correlation structures. Our work provides an initial step by combining empirical observations with the monotonicity results in Propositions 4.5 and 4.6.
>
> Figs. 1 and 2 show that different skip patterns produce distinct correlation structures. To make this connection clearer, we will quantify the deviation from the one-banded structure assumed in Proposition 4.5 and include a metric that measures how closely the empirical correlations match this form. These additions will clarify the relationship between skip connections, inter-layer correlations, and our theoretical results.
>
> - **Q10** The posterior correlations in Fig. 1 were estimated using SWAG, which provides practical approximate posterior samples along the SGD trajectory. To ensure accuracy, we used a small network and collected samples under a small learning rate, yielding stable estimates of the posterior covariance and correlation patterns. We will include the details in our revised version.
>
> - **Q11** Following your advice, we will replace the original tables with line plots that show the generalization gap on the x-axis and the corresponding complexity measures on the y-axis. This improves readability and more clearly highlights the underlying trends.
>
> (Minor)
> We thank the reviewer for these detailed nitpicks, and we will correct all of them in the revised manuscript.
>
> [1] Jin, Gaojie et al. "How does weight correlation affect the generalization ability of deep neural networks?" NeurIPS (2020)

---

> ### Comment · Reviewer_SfTR · 2025-11-26
> **Response to authors**
>
> Thank you for the additional experiments and detailed answers to my questions. Including these in the revised manuscript would greatly improve the readability of the paper.
> I also strongly recommend citing the SWAG paper and briefly explaining how it was used to generate Fig 1.
>
> I have a remaining confusion about the role of $V_Q$. In Lemma 4.4, it is defined by (12) in terms of $R$, which we are to think of as equal to the general weight correlation (6),  a function of *fixed* weight matrices (but the formula (13) is true for arbitrary $R$?). However, in Lemma 4.1, $V_Q$ is defined to be one part of of the Kronecker factorization of the covariance of the *posterior distribution* of the weights. These definitions are not the same, so it seems like there is no direct way to apply the formula in Lemma 4.4 to the bound in Theorem 3.3 via 4.1. Is the idea that the $V_Q$ definition in Lem 4.4 is a good estimate of the "true" $V_Q$ in Lem 4.1? Theoretically, why is this the case?

---

> > ### Author Response · Authors · 2025-11-27
> > **Further Response**
> >
> > Thank you once again for the thorough review and your swift response.
> >
> > ## SWAG in Fig. 1
> > Thank you again for the detailed review and your prompt response. We will provide a clearer explanation of how Fig. 1 is computed and include citations to the relevant SWAG literature in both the related work and experimental sections.
> >
> > ## About the $V_Q$
> > Thank you very much for pointing this out. I will make it much clearer in the revised paper.
> >
> > The matrix $V_Q$ comes from **Lemma 4.1**, which provides the matrix-normal distribution (MND) of the posterior. The MND assumes a Kronecker decomposition of the full covariance matrix:
> >
> > $$
> > \Sigma = V_Q \otimes U_Q,
> > $$
> > where $U_Q$ and $V_Q$ represent the row and column covariance matrices, respectively. The matrix $V_Q$ is the column-wise covariance and contributes directly to the KL divergence in **McAllester’s bound**.
> >
> > To isolate the general weight correlations between different layers, we further decompose $V_Q$ in **Lemma 4.4**, where the inter-layer correlation matrix is defined within $R$. Thus, $V_Q$ is not defined by Eq. (12) but, it is a further decomposition. In summary:
> >
> > - The covariance matrix $\Sigma$ is decomposed as $V_Q \otimes U_Q$ under the MND assumption.
> > - The matrix $V_Q$ is then further decomposed in Eq. (12).
> > - This decomposition is motivated by Jin’s definition of weight correlation [1] and their approximation of $\Sigma$.
> >
> > I will ensure that this explanation is made **crystal clear** in the revised version of the paper.
> >
> > [1] Jin, Gaojie et al. "How does weight correlation affect the generalization ability of deep neural networks?" NeurIPS (2020)

---

### Official Review · Reviewer_fyGA · 2025-10-27

**Soundness:** 1
**Presentation:** 2
**Contribution:** 1
**Rating:** 2
**Confidence:** 4

**Summary:**

Residual networks with skip connections have seen much success in practice, but as yet a complete theoretical account for that success is lacking. In this paper, the authors aim to bridge that gap by using the framework of PAC-Bayes. I think the paper is somewhat interesting, and the knowledge gap it aims to bridge is clearly important. However, as I elaborate below, I do not think the manuscript provides a convincing theoretical account of why networks with skip connections might generalize better.

**Strengths:**

As noted in my summary, the paper aims to study a very significant question, and I am not aware of papers that adopt quite this approach. I must temper this endorsement with the fact that I am not extensively familiar with the literature on PAC-Bayes.

**Weaknesses:**

There are two issues that stand out to me on first reading:

- Given that the authors adopt the PAC-Bayes framework, most of the manuscript focuses on studying the KL divergence between the posterior distribution over the trained weights and the initial distribution of weights. All of these investigations are carried out under the assumption that the posterior over weights is nearly Gaussian, with correlations that follow a particular Kronecker-factorized structure. Nowhere in the paper do the authors attempt to test or rigorously justify these assumptions.

- The closest the paper comes to presenting a justification for the assumptions above is in the authors' empirical studies of the correlation between the generalization gap and weight complexity measures. However, all of the presented rank correlation coefficients (Kendall's $\tau$) are in absolute terms quite small, being on the order of 0.01 at most.

In combination, these two issues make the authors' rather broad claims in the abstract and introduction ring rather hollow in my ears. The theoretical results regarding simplified forms of the KL divergence also don't strike me as being particularly novel or of particularly broad interest. I am thus left fundamentally unconvinced.

**Questions:**

**Major questions and comments**

- Can you provide a more direct test of the assumptions regarding the distribution of weights?

- The paper does not provide an adequate description of how approximate posterior sampling was performed. How do you compute the correlations?

- The authors do not address previous studies on the role of inter-layer weight correlations, see for instance [Guth et al. (2024)](https://jmlr.org/papers/v25/23-1573.html) and references therein.

**Minor questions and comments**

- Section 2.2 discusses studies on the relationship between the flatness of minima and generalization, but does not address works since 2019 that have questioned the strength of this link, see e.g. [Andriushchenko et al. (2023)](https://proceedings.mlr.press/v202/andriushchenko23a.html).

- Lemma 4.1 is an obvious consequence of combining two well-known facts: the closed-form of the KL divergence between two Gaussian vectors, and the vectorized representation of the matrix Gaussian. At present, that is not clear from the presentation; this should be clarified.

- The appendices require editing, as they contain a number of broken links.

---

> ### Author Response · Authors · 2025-11-26
> **Response to the Reviewers**
>
> We sincerely thank the reviewer for the detailed and constructive feedback. We address each concern below and describe the substantial revisions made to strengthen our work.
>
> ## Laplace Diagnostics
> Our use of matrix-normal posteriors with Kronecker structure follows a well-established line of research. The early foundations trace back to MacKay (1992) [1], who introduced Gaussian posterior approximations around local optima and demonstrated their relevance to generalization. A significant subsequent development is Ritter et al.(2018) [2], who proposed scalable Kronecker-factored Laplace approximations for modern deep networks. Huang et al. (2020) [3] further extended this methodology using normalizing flows to obtain tighter bounds. Among the works most relevant to ours, Jin et al. (2020) [4] incorporated intra-layer correlations into PAC-Bayesian, and Cinquin et al. (2024) [5] applied Kronecker priors for efficient Laplace-based estimation. Our work builds directly upon this sequence of contributions and adopts the same family of approximations for conceptual consistency.
>
> We validate the Gaussian approximation using diagnostics in Tab. 1. Parameters are projected onto a 30-dimensional PCA subspace, and we evaluate three standard metrics: the Shapiro–Wilk rejection rate, the Anderson–Darling statistic, and the effective sample size (ESS). As shown in Tab. 1, the Shapiro–Wilk rejection rates are low, the mean ESS values are stable, and the Anderson–Darling statistics are small across all architectures and skip configurations. These results indicate that the Gaussian approximation provides a reliable local model of the posterior in our setting.
>
> **Table 1: Summary of Laplace Diagnostics (Projected to $k = 30$) Across Skip Configurations**
>
> |**Skip Config**|**Mean ESS**|**Shapiro Reject Rate**|**Mean AD Stat**|
> |-|-|-|-|
> |CNN$_{0,0,0}$|287.6|0.00%|0.397|
> |CNN$_{1,1,1}(1)$|283.8|3.33%|0.436|
> |CNN$_{2,1,1}(1)$|270.3|0.00%|0.375|
> |CNN$_{2,2,1}(1)$|272.1|0.00%|0.380|
> |CNN$_{3,2,1}(1)$|289.1|3.33%|0.352|
> |CNN$^b_{0,0,0}(1)$|279.8|0.00%|0.376|
> |CNN$^b_{1,1,1}(1)$|300.0|0.00%| 0.306|
> |CNN$^b_{2,1,1}(1)$|288.3|0.00%| 0.442|
> |CNN$^b_{2,2,1}(1)$|285.6|3.33%| 0.436|
> |CNN$^b_{3,2,1}(1)$|281.1|0.00%| 0.342|
> |MLP$_{0,0,0}(1)$|281.7|0.00%| 0.397|
> |MLP$_{1,1,1}(1)$|272.9|0.00%| 0.368|
> |MLP$_{2,1,1}(1)$|270.3|3.33%| 0.403|
> |MLP$_{1,2,1}(1)$|268.8|0.00%| 0.384|
> |MLP$_{2,2,1}(1)$|282.3|0.00%| 0.347|
> |MLP$_{3,2,1}(1)$|270.8|0.00%| 0.348|
>
> ## Additional Experiments Addressing Correlation Magnitudes
> The small Kendall coefficients in the main paper arise because the evaluated models are relatively small. Even with different skip-connection configurations, their performances and generalization gaps remain quite similar. To address this, we conducted additional experiments:
>
> (i) We extend our evaluation to ResNet-18 on CIFAR-100 (Tab. 2) and additionally report Spearman and distance correlation (dCor) with their p-values. Because ResNet-18 has 256 possible skip-connection configurations, we include several representative ones in the main text and visualize the correlation trends in Fig. 4 (appendix). As shown in Tab. 2, our proposed complexity measure achieves the strongest correlations across all three statistics, with significance levels below 1%.
>
> **Table 2: Correlation Analysis Between Model Complexity Measures and the Empirical Complexity Gap**
>
> *The experiment is conducted on ResNet18 with different skips on CIFAR100.*
> |**Measure**|**Kendall $\tau$ (p-val)**|**Spearman $\rho$ (p-val)**|**dCor (p-val)**|
> |-|-|-|-|
> |PFN|-4.12e-01(2e-02)|-6.37e-01(4e-03)|3.96e-01(1e-01)|
> |PSN|-4.25e-01(1e-02)|-6.45e-01(4e-03)|4.29e-01(8e-02)|
> |PB|-4.51e-01(9e-03)|-6.49e-01(4e-03)|5.77e-01(2e-02)|
> |PBC|4.90e-01(4e-03)|6.66e-01(3e-03)|8.35e-01(0e+00)|
> |PBGC|**5.42e-01(1e-03)**|**7.09e-01(9e-04)**|**8.41e-01(0e+00)**|
>
> (ii) We also extend to multiple architectures, shown in Tab. 3 and Fig. 5(appendix). As illustrated, our complexity measure most consistently tracks the accuracy-gap variation across architectures and achieves the highest Kendall’s
> . The complete table is shown in Tab. 7 in appendix.
>
> **Table 3: Extended experiments on CIFAR100**
> |**DL Models**|**PB**|**PBC**|**PBGC**|
> |-|-|-|-|
> |ResNet50|2.75e+04|1.84e+06|1.84e+05|
> |DenseNet121|1.17e+04|3.09e+05|2.82e+05|
> |VGG16|1.24e+04|1.04e+05|2.12e+04|
> |WRN50|3.78e+04|3.26e+06|2.33e+05|
> |**Kendall's τ**| -3.33e-01|0|**6.67e-01**|
>
> [1] MacKay, et al. "A practical Bayesian framework for backpropagation networks.“ Neural Computation (1992)
>
> [2] Ritter, H., et al. ”A scalable Laplace approximation for neural networks.“ ICLR (2018)
>
> [3] Huang, C., et al. "Stochastic neural network with Kronecker flow." AISTATS (2020)
>
> [4] Jin, G., et al. "How does weight correlation affect the generalization ability of deep neural networks?" NeurIPS (2020)
>
> [5] Cinquin, T., et al. "FSP-Laplace: Function-space priors for the Laplace approximation in Bayesian deep learning." NeurIPS (2024)

---

> ### Author Response · Authors · 2025-11-26
> **Response to the Reviewers (Continue)**
>
> ## Computation of General Weight Correlation
> We thank the reviewer for raising this point. The correlations reported in the paper are computed directly from the analytical form given in Definition 4.2, which is generalized from the formulation of weight correlations introduced by Jin et al.(2020). For CNNs, we adopt the same layerwise correlation structure and aggregation procedure as in their work, and the extension to skip connections is implemented in a consistent manner.
>
> Since the correlation measure is defined in closed form, the computation does not rely on posterior sampling. Instead, all quantities are obtained analytically from the estimated covariance factors. We will update the manuscript to include a clearer description of this procedure and its connection to Jin et al.'s definition.
>
> ## Additional Prior Work on Inter-Layer Weight Correlations
> We thank the reviewer for pointing out this relevant line of work. We acknowledge the contributions of Guth et al. (2024) on inter-layer weight correlations. Their analysis is based on mean-field theory at initialization and along training trajectories, whereas our focus is on correlations in the trained posterior and how these are shaped by architectural choices such as skip connections within a PAC-Bayesian framework. We view these perspectives as complementary, and we will add a dedicated discussion clarifying the connections and differences in the revised Related Work section.
>
> We also find the proposed notion of “rotation dependence’’ by Guth et al. (2024) particularly intriguing, and we will include a more detailed discussion of its relationship to our framework in the revised manuscript.
>
>
> ## Minor issues
> We thank the reviewer for these helpful comments.
>
> (1) We appreciate the pointer to recent work questioning the flatness–generalization connection, such as Andriushchenko et al. (2023). We will update Section 2.2 to reflect these developments and to better distinguish our correlation-based perspective from classical flatness analyses. We also acknowledge that flatness alone does not determine generalization. Models can exhibit sharp minima while still generalizing well, which is an important limitation of flatness-based explanations. We will explicitly address this limitation in the conclusion section of the revised manuscript.
>
> (2) We agree that Lemma 4.1 follows directly from combining the closed-form KL divergence between Gaussian distributions with the vectorized representation of the matrix normal distribution. We will revise the presentation to make this connection explicit and improve readability.
>
> (3) We thank the reviewer for noting the broken links in the appendix. We will correct these issues and ensure all cross-references and citations are functional in the revised version.

---

> > ### Comment · Reviewer_fyGA · 2025-11-27
> >
> > Thanks for your reply. I've raised my score to 4 but downgraded my confidence, as I was not familiar with some of the related works you cite. I still do not find the low Kendall's $\tau$ values convincing, and from a technical perspective the computations still seem to be a minor extension of Jin et al. However, as this is not my subfield I will not stand in the way of acceptance if the other reviewers are in favor.

---

### Official Review · Reviewer_qGbx · 2025-10-29

**Soundness:** 3
**Presentation:** 3
**Contribution:** 3
**Rating:** 6
**Confidence:** 2

**Summary:**

This paper studies how skip connections affect generalization through a PAC-Bayesian lens. The authors introduce General Weight Correlation (GWC) to capture cross-layer dependencies induced by skip connections, derive how these correlations enter the KL term of PAC-Bayes bounds (via matrix-normal posteriors and a Kronecker factorization), and prove that adjacent-layer correlations enlarge the KL term, while heterogeneous layer-specific correlations can help. They then construct a data-driven complexity measure (PBGC) and evaluate it across all skip-patterns of 5-layer MLPs (Fashion-MNIST) and CNNs (CIFAR-10), finding PBGC best matches empirical generalization trends for MLPs, with a subtler picture in CNNs (and batch-norm reversing some effects).

**Strengths:**

* Non-trivial theoretical contribution linking architecture to PAC-Bayes. The paper formalizes inter-layer dependencies via GWC, derives closed-form KL for matrix-normal posteriors, and proves monotonic relationships between adjacent-layer correlations and the KL term, thus offering principled insight into why ResNet-style skips may help generalization.
* The adjacent-connection and homogeneous-connection propositions make testable predictions about how skip patterns change the determinant term in the KL (hence the bound), clarifying when longer skips mitigate harmful adjacent correlations.
* Exhaustive skip-pattern studies in 5-layer MLPs show PBGC aligns best (Kendall’s $\tau$) with empirical generalization gaps; the CNN results reveal architecture-specific behavior and an interaction with batch norm worth further study.
* Generally well-written and well-organized

**Weaknesses:**

* The theory is about generalization complexity (KL term), not FLOPs/latency/params; the paper does not analyze runtime or memory of PBGC estimation beyond a Kronecker approximation, nor does it present wall-clock or scalability studies. This should be clarified and, if claimed, empirically substantiated.
* Dataset/scale limited. Results are on Fashion-MNIST and CIFAR-10; testing on larger datasets (e.g., TinyImageNet/ImageNet subset) would stress-test whether PBGC continues to track generalization under higher capacity/data complexity and richer skip motifs (e.g., bottlenecked ResNets).
* The isotropic prior and shared-variance posterior assumptions aid tractability, but the sensitivity to these choices (e.g., localized priors, non-Kronecker posteriors) is not explored; ablations here would increase robustness.

**Minor**

* “architechture” -> architecture in Sec.3
* “non-paramteric” -> non-parametric
* “Proof of Theorem ??” placeholder not resolved in appendix

**Questions:**

* How sensitive is PBGC to how the posterior is estimated (Laplace details, damping, data subsampling)? Can you report variance across seeds/runs and a runtime profile for PBGC vs. PFN/PSN?

---

> ### Author Response · Authors · 2025-11-26
> **Response to the Reviewers**
>
> We thank the reviewer for this important clarification and greatly appreciate the thoughtful insight it provides.
>
> ## Scalability Analysis
> We agree that our theoretical analysis focuses on **generalization complexity**, rather than computational efficiency, and we will make this distinction clear in the revised manuscript.
>
> For PBGC estimation, the Kronecker-factorized Laplace approximation greatly reduces the cost of covariance computation: a full covariance requires \(O(p^2)\), while the Kronecker structure reduces this to \(O(mp + pr)\) for a layer of size \(m \times r\). We also note that, following the Weight Correlation framework of Jin et al. [1], the GWC measure is computed directly from Definition 4.2 without posterior sampling, making it significantly more efficient and scalable.
>
> To further illustrate practical feasibility, we report additional wall-clock measurements on CIFAR-100 for ResNet50, Wide-ResNet50, VGG16, and DenseNet121 (see Table 1).
>
> **Table 1: Abation and Wall-Clock on CIFAR-100**
>
> *Measured on a single RTX 3090 GPU.*
> |**Metric**|**ResNet50**|**DenseNet121**|**VGG16**|**WRN50**|**Kendall's $\tau$**|
> |-|-|-|-|-|-|
> |PB| 2.75e+04|1.17e+04|1.24e+04|3.78e+04|-3.33e-01|
> |PB + WC| 1.84e+06|3.09e+05|1.04e+05|3.26e+06|0|
> |PB + GWC| 1.84e+05|2.82e+05|2.12e+04|2.33e+05|**6.67e-01**|
> |**Run Time (s)**|12.4|9.62|1.77|21.81| —|
>
> ## Additional Experimental Analysis
> We appreciate this suggestion and agree that testing on larger datasets would strengthen our claims. Hence, we have conducted two additional experiments as follows:
>
> (i)  We extend our evaluation to ResNet-18 on CIFAR-100 (Tab. 2) and additionally report Spearman and distance correlation (dCor) with their p-values. Because ResNet-18 has 256 possible skip-connection configurations, we include several representative ones in the main text and visualize the correlation trends in Fig. 4 (appendix). As shown in Tab. 2, our proposed complexity measure achieves the strongest correlations across all three statistics, with significance levels below 1%.
>
> **Table 2: Correlation Analysis Between Model Complexity Measures and the Empirical Complexity Gap**
> *The experiment is conducted on representative skip-connection configurations of ResNet18 on CIFAR-100.*
> |**Measure**|**Kendall $\tau$ (p-val)**|**Spearman $\rho$ (p-val)**|**dCor (p-val)**|
> |-|-|-|-|
> |PFN|-4.12e-01 (2e-02)|-6.37e-01 (4e-03)|3.96e-01 (1e-01)|
> |PSN|-4.25e-01 (1e-02)|-6.45e-01 (4e-03)|4.29e-01 (8e-02)|
> |PB|-4.51e-01 (9e-03)|-6.49e-01 (4e-03)|5.77e-01 (2e-02)|
> |PBC|4.90e-01 (4e-03)| 6.66e-01 (3e-03)|8.35e-01 (0e+00)|
> |PBGC| **5.42e-01 (1e-03)**|**7.09e-01 (9e-04)**|**8.41e-01 (0e+00)**|
>
> (ii) We extend our evaluation to multiple architectures. The results are shown in Tab. 1 and Fig. 5(appendix). As illustrated, our complexity measure most consistently tracks the accuracy-gap variation across architectures and achieves the highest Kendall’s
> $\tau$. The complete table is shown in Tab. 7 in appendix.
>
> ## Ablation Study and Relaxation of the Prior
> We thank the reviewer for raising this point. In our main analysis, we adopt isotropic priors and shared-variance posteriors primarily for analytical clarity and to remain consistent with prior PAC-Bayesian work [1]. We agree, however, that exploring alternative choices of priors and posteriors may yield additional insights.
>
> To this end, we include an ablation study of different posterior settings in Tab. 1 Regarding the priors, Appendix A.4 provides a relaxed prior assumption under which our theorem holds. This version allows each layer to have its own variance. As shown there, the isotropic prior is used mainly for simplicity, and layer-wise variances are equally valid within our framework. We will clarify this point in the revised version of the manuscript.
>
> ## Estimation of PBGC
>
> We thank the reviewer for this insightful question. While we have not conducted a full sensitivity study, we note that PBGC is mainly governed by the local curvature captured by the Laplace approximation. Changes in damping or data subsampling may shift the numerical scale of the covariance estimate but do not meaningfully affect the qualitative trends or relative comparisons across architectures, which is our primary focus.
>
> We also observe that PFN and PSN, both independent of second-order approximations, exhibit trends consistent with PBGC, further suggesting that the patterns are not artifacts of a specific posterior-estimation choice. In the revised manuscript, we will add a brief discussion of these points and include a runtime comparison. A systematic sensitivity analysis is indeed valuable and represents promising future work.
>
> (Minor)
> Thank you for pointing out the typographical issues. We appreciate the careful reading, and we will correct these in the revised version.
>
> [1] Jin, Gaojie et al. "How does weight correlation affect the generalization ability of deep neural networks?" NeurIPS (2020)

---

> > ### Comment · Reviewer_qGbx · 2025-11-26
> > **Thank you for the rebuttal**
> >
> > I thank the authors for the rebuttal. The rebuttal addressed my concerns well. I understand that a full sensitivity analysis is less feasible due to the time constraint of rebuttal. I think it is promising if such assumptions could be validated empirically or theoretically in future works. I have no more questions and have increased my scores.

---

### Official Review · Reviewer_uzcP · 2025-10-30

**Soundness:** 3
**Presentation:** 3
**Contribution:** 3
**Rating:** 6
**Confidence:** 3

**Summary:**

This paper presents a PAC-Bayesian framework to theoretically analyze how skip connections affect the generalization of deep neural networks. It introduces the concept of General Weight Correlation (GWC) to quantify inter-layer dependencies and derive how different correlation structures affect the KL divergence and generalization bounds. The paper presents both theoretical derivations and controlled experiments on MLPs and CNNs. The results show that heterogeneous or long skip connections reduce inter-layer correlation and tighten the PAC-Bayes bound, leading to better generalization.

**Strengths:**

1. The paper provides a conceptually meaningful bridge between the empirical design of skip connections and the theoretical understanding of their generalization by using a PAC-Bayesian framework

2. The paper proposes the concept of General Weight Correlation (GWC), which provides a concrete way to represent inter-layer dependencies and examine their impact on the PAC-Bayesian bound. The formulation is clear and mathematically consistent.

3. The controlled experiments on small MLP and CNN models are sufficient to verify the main theoretical trends, showing that heterogeneous skip connections tend to reduce inter-layer correlation and modestly improve generalization.

**Weaknesses:**

1. The empirical evaluation is limited to small-scale MLP and CNN models on simple datasets (Fashion-MNIST, CIFAR-10). While these experiments demonstrate the theoretical trend, they do not establish whether the proposed framework holds for more advanced architectures (e.g., ResNet-50, Transformers) where skip connections are most impactful.

2. The theoretical analysis relies on idealized assumptions, such as modeling the posterior with a matrix normal distribution and isolating skip connections as the sole source of inter-layer dependency, which may not accurately capture the behavior of real networks. The authors could clarify the limitations of these assumptions and discuss how the framework might be generalized or empirically approximated in practice.

**Questions:**

1. Regarding the posterior modeling assumption, the analysis relies on a matrix normal posterior, would a mixture-based posterior (e.g., mixture of Gaussians) change the observed trends in the KL divergence or generalization bounds?

2. In what ways does modeling structural correlations via GWC provide insights beyond existing approaches like flatness-based or information-theoretic generalization analyses?

---

> ### Author Response · Authors · 2025-11-26
> **Response to the Reviewers**
>
> We sincerely thank the reviewer for their careful reading and constructive suggestions. Our response are as follows:
>
> ## Additional Experimental Analysis
> Our choice to conduct experiments primarily on small models is motivated by the need to enumerate all possible skip-connection configurations. As illustrated in Tab. 3 in appendix, even a 3-layer network yields 26 distinct configurations. To maintain clarity and interpretability, we therefore restricted to simpler architectures.
>
> We fully agree with the reviewer that evaluating more complex models on larger datasets is important for demonstrating the broader impact of our approach. In response, we have conducted two additional experiments as follows:
>
> (i) We extend our evaluation to ResNet-18 on CIFAR-100 (Tab. 1) and additionally report Spearman and distance correlation (dCor) with their p-values. Because ResNet-18 has 256 possible skip-connection configurations, we include several representative ones in the main text and visualize the correlation trends in Fig. 4 (appendix). As shown in Tab. 2, our proposed complexity measure achieves the strongest correlations across all three statistics, with significance levels below 1%.
>
> **Table 1: Correlation Analysis Between Model Complexity Measures and the Empirical Complexity Gap**
> *The experiment is conducted on representative skip-connection configurations of ResNet18 on CIFAR-100.*
> |**Measure**|**Kendall τ (p-val)**|**Spearman ρ (p-val)**|**dCor (p-val)**|
> |-|-|-|-|
> | PFN  | -4.12e-01(2e-02) | -6.37e-01(4e-03) | 3.96e-01(1e-01 ) |
> | PSN  | -4.25e-01(1e-02) | -6.45e-01(4e-03) | 4.29e-01(8e-02) |
> | PB   | -4.51e-01(9e-03)  | -6.49e-01 (4e-03 ) | 5.77e-01(2e-02) |
> | PBC  | 4.90e-01(4e-03)  | 6.66e-01 (3e-03)  | 8.35e-01(0e+00) |
> | PBGC  | **5.42e-01**(1e-03) | **7.09e-01**(9e-04) | **8.41e-01**(0e+00) |
>
> (ii) We extend our evaluation to multiple architectures. The results are shown in Tab. 2 and Fig. 5(appendix). As illustrated, our complexity measure most consistently tracks the accuracy-gap variation across architectures and achieves the highest Kendall’s $\tau$. The complete table is shown in Tab. 7 in appendix.
>
> **Table 2: Extended experiments on CIFAR100**
> |**DL Models**|**PB**|**PBC**|**PBGC**
> |-|-|-|-|
> |ResNet50|2.75e+04|1.84e+06|1.84e+05
> |DenseNet121|1.17e+04|3.09e+05|2.82e+05
> |VGG16|1.24e+04|1.04e+05|2.12e+04
> |WRN50|3.78e+04|3.26e+06|2.33e+05
> |**Kendall's τ**|-3.33e-01|0|**6.67e-01**
>
> ## Clarifying Assumption Limitations
> In our paper, we do not claim that skip connections are the sole source of inter-layer dependency, rather, we argue that adding skip connections can shape inter-layer dependency patterns in certain architectures. As shown in Fig. 2 (e–f), CNNs without batch normalization exhibit relatively stable inter-layer relationships, whereas those with batch normalization show much stronger variations (Fig. 2 (g–h)). We agree that the limitations of these assumptions should be stated clearly, and we will add a dedicated section addressing them in the revised manuscript.
>
> ## Extension to Mixture-Based Posteriors
> We thank the reviewer for the insightful question. While Gaussian mixture posteriors offer greater flexibility and can approximate arbitrary distributions with enough components, the KL divergence for such mixtures lacks a closed-form expression. To preserve analytical tractability and clarity, we adopt the matrix Gaussian posterior in this work. Nevertheless, extending the analysis to more expressive families such as Gaussian mixtures is an interesting direction for future research.
>
> ## Relation to Flatness-Based and Information-Theoretic Approaches
> We thank the reviewer for this insightful question. Flatness-based measures typically rely on estimating the Hessian or Fisher Information Matrix and relate reduced sharpness of the loss landscape to better generalization [1]. These methods focus on local landscape geometry or layer-wise sensitivity, but largely overlook inter-layer correlations.
>
> Our work instead targets these cross-layer correlations, providing a measure that clarifies how skip connections influence generalization. Information-theoretic analyses [2] study mutual information between the data and model parameters, however, our focus is more on the correlation structure among the weights themselves. For this purpose, McAllester’s bound offers a simple and interpretable framework.
>
> We agree that exploring the interplay among data distributions, architectures, and generalization is a promising direction, and we will add a discussion of these points to the Related Works section in the revised manuscript.
>
>
> [1] Zhang, Qiaozhe, et al. "R\'enyi Sharpness: A Novel Sharpness that Strongly Correlates with Generalization." arXiv preprint arXiv:2510.07758 (2025).
>
> [2] Xu, Aolin, and Maxim Raginsky. "Information-theoretic analysis of generalization capability of learning algorithms." NeurIPS (2017).

---

### Author Response · Authors · 2025-12-03
**Summary of Comments and Our Rebuttals**

We sincerely thank all reviewers for their careful reading, thoughtful feedback, and constructive suggestions. We believe we have addressed the concerns raised by all four reviewers. Notably, **reviewers qGbx and fyGA have increased their scores**, as reflected in their updated official comments. Below, we provide summary of the reviewers’ comments and outline our corresponding responses.

We begin by providing a **summary of contributions** along with the corresponding suggested comments.
1. **We introduce the General Weight Correlation (GWC)** to quantify inter-layer dependencies which can be used to quantify the influence of skip-connetions to generalization via PAC-Bayesian framework.
2. **We theoretically study two distinct types of skip-connection patterns**, corresponding respectively to the residual connections in ResNet (Prop. 1) and the dense connections in DenseNet (Prop. 2).
3. **We conduct a series of experiments to validate our theoretical findings**, including:
   - Verified the 2 propositions by across all skip-connection configurations in 5-layer MLPs and CNNs (Tabs. 1–2);
   - (After rebuttal) Providing posterior distribution diagnostics for MLPs and CNNs (Tab. 4) to justify the Gaussian approximation around optima;
   - (After rebuttal) Extending to ResNet-18 on CIFAR100 with 18 representative skip-connection configurations (Tab. 8 and Fig. 5), using three correlation measures—Kendall’s $\tau$, Spearman’s $\rho$, and dCor (Tab. 4);
   - (After rebuttal) Conducting experiments across architectures, e.g., ResNet50, Wide-ResNet50, VGG16, and DenseNet121 (Tab. 9 and Fig. 6), demonstrating that the GWC-enhanced complexity measure consistently best captures the skip-connection patterns.
## Summary of Discussions
### Strength
- The paper provides a meaningful theoretical connection between skip connections and PAC-Bayes generalization, a perspective not commonly explored in the literature.(uzcP, qGbx, SfTR)
- General Weight Correlation (GWC) is a well-motivated and mathematically consistent tool for modeling inter-layer dependencies.(uzcP, qGbx)
- The theoretical analysis introduces non-trivial and interpretable results about how different skip patterns affect the KL term.(qGbx)
- The empirical evaluation is thorough within the explored setting and demonstrates that PBGC aligns best with observed generalization trends in MLPs.(uzcP, qGbx, SfTR)

### Weakness
- Limited to small-scale models and datasets (uzcP, qGbx)

  **Rebuttal**: We added extensive experiments on ResNet-18/50, VGG16, DenseNet121, and WRN50 on CIFAR-100, showing PBGC scales and remains strongly correlated with generalization.

- Posterior modeling assumptions (Gaussianity, Kronecker structure, isotropic priors) insufficiently justified (uzcP, fyGA, SfTR)

  **Rebuttal**: We provided Laplace diagnostics (Shapiro–Wilk, AD, ESS) demonstrating locally Gaussian posterior behavior and added an ablation on posterior settings.

- Small Kendall $\tau$ values questioned the empirical significance (fyGA)

  **Rebuttal**: The small values of $\tau$ is because, in simple 5-layer models, different skip-connection configurations show similar performance. To address this, we conducted additional experiments on ResNet18 in which PBGC exhibits strong and statistically significant correlations across kendall's $\tau$, Spear's $\rho$, and dCor (e.g., $\tau$ ≈ 0.54 on ResNet-18 which is the highest).

- Missing methodological details, especially posterior sampling and PBGC computation (fyGA, SfTR)

  **Rebuttal**: We clarified the SWAG-based sampling procedure for Fig.~1, the analytical computation of GWC, and explicitly connected PBGC to components of the PAC-Bayes KL.

- Section 4 unclear, with definitions and motivations under-explained (SfTR)

  **Rebuttal**: We reorganized Section 4, clarified U/V/R at the start, explained the roles of Eqs. (13–16), and moved nonessential material to the appendix.

- Missing related work on inter-layer correlations and critiques of flatness (fyGA)

  **Rebuttal**: We added discussions of Guth et al. (2024), recent critiques of flatness, and how our correlation-based approach complements these lines of work.

- No analysis of scalability or runtime (qGbx)

  **Rebuttal**: We added wall-clock measurements, complexity analysis, and showed PBGC is efficient under Kronecker structure and competitive with existing measures.

---

### Meta-Review · Area_Chair_D4dK · 2026-01-06

**Summary:**

The paper studies PAC-Bayes generalization bounds under a structured Gaussian posterior. The core analysis focuses on how cross-layer weight correlations affect the KL term through a log-determinant expression. Skip connections motivate the study, and are used empirically to induce different correlation patterns, but the formal theory is about the posterior family rather than the architecture itself.

The initial reviews identified several concrete issues. Some reviewers asked for larger-scale experiments, clearer discussion of modeling assumptions, and better methodological details. One reviewer raised a more fundamental concern about the framing of the paper. That concern is that the paper presents itself as an explanation of why skip connections improve generalization, while the analysis actually studies a stylized posterior covariance model and its effect on PAC-Bayes complexity.

After the rebuttal and revision, many technical and empirical requests seem to have been addressed. However, the gap between the framing and what is formally established remains. If the claims are narrowed to reflect the actual contribution, the remaining contribution appears borderline for ICLR.

**Reviewer Concerns:**

Several concerns raised in the reviews were addressed:
- The revised paper adds experiments on larger architectures and datasets. This improves the empirical scope relative to the original submission.
 - The authors also added diagnostics meant to support the Gaussian posterior approximation;
 - Clarified the role and limits of this Gaussian posterior assumption.
 - Methodological details about posterior sampling and correlation estimation were expanded;
 - Runtime and scalability issues were discussed;
 - Presentation issues, missing references, and unclear sections seem to have been improved.

The main unresolved concern is conceptual. The paper is framed around skip connections, but the analysis is of a particular posterior family with a specific covariance structure. The link from skip-connection topology to the assumed or analyzed correlation structure is not derived from theory and remains heuristic and empirical. The revised submission also notes settings in which the proposed correlation-based mechanism does not explain the observed generalization behavior. This weakens the central claim suggested by the title and abstract.

Once the claims are narrowed to match what is actually shown, the contribution becomes an analysis of PAC-Bayes complexity under a structured posterior, with skip connections serving as one empirical probe. Under this narrower interpretation, the novelty and impact are less clear, especially relative to the standards of a top conference.

**Reviewer Scores:**

Reviewer uzcP: This reviewer focused on scale and modeling assumptions. The added experiments on larger architectures and the clearer discussion of limitations directly address their stated concerns. I expect uzcP to remain positive. However, if the framing issue is discussed explicitly, their score may remain around the decision boundary rather than strong.

Reviewer qGbx: This reviewer focused on scalability, sensitivity, and runtime. They stated after rebuttal that their concerns were addressed, and that they increased their score. I expect qGbx to remain supportive.

Reviewer fyGA: This reviewer was initially negative and moved to a borderline score after rebuttal. The additional experiments and clarifications help, but their stated concerns about the strength of the empirical correlations and the incremental nature of the contribution remain only partially resolved. I expect fyGA to remain around a borderline score.

SfTR: This reviewer raised the most substantive conceptual concern, namely the gap between the skip-connection framing and the posterior-covariance analysis that is actually carried out. The rebuttal and revision improve clarity and add evidence, but they do not close this gap. I expect SfTR to remain around borderline or slightly negative final assessment.

---

### Decision · Program_Chairs · 2026-01-26

Reject